Review Article

EMBO
Molecular Medicine

# Hypothalamic regulation of sepsis-associated anorexia: cytokine and hormonal signalling through AgRP/POMC circuits

Wanting Zhu[1,2], Claude Libert (ID)[1,2] & Tineke Vanderhaeghen (ID)[1,2] ✉

## Abstract

Sepsis is a life-threatening syndrome resulting from a dysregulated host response to an infection and is considered as a major global health priority. Despite increased metabolic energy needs to fight the infection and to sustain the inflammatory response, anorexia is one of the main characteristics of sickness behaviour during sepsis. In this review, we address the question of how feeding behaviour is regulated under basal conditions at the level of the hypothalamus, with specific focus on the orexigenic agouti-related peptide (AgRP)/neuropeptide Y (NPY)-expressing neurons and the anorexigenic pro-opiomelanocortin (POMC)/cocaine and amphetamine-regulated transcript (CART)-expressing neurons present in the arcuate nucleus. This is mediated by neural and humoral pathways involving the vagal nerve, and the blood-brain barrier and circumventricular organs, respectively. Furthermore, we discuss recent advances in how sepsis affects these appetite-controlling mechanisms by impairing the central integration of these peripheral signals and suggest potential therapeutic targets that might prevent or revert sepsis-associated anorexia.

**Subject Categories** Immunology; Microbiology, Virology & Host Pathogen Interaction; Neuroscience

## Introduction

Sepsis is a life-threatening disease characterised by acute organ dysfunction and associated with high mortality (Cecconi et al, 2018). With an annual incidence of 49 million cases and 11 million deaths, the World Health Organization has recognised sepsis as a global health priority (Reinhart et al, 2017; Rudd et al, 2020). Despite decades of intensive research, no successful therapeutic drug has hit the clinical market that clearly improves the patient outcome (Van Wyngene et al, 2018). Sepsis pathophysiology is characterised by an initial hyperinflammatory phase that lasts for a few days, followed by an immunosuppressive phase. Nevertheless,

clinical trials focusing on immune-stimulating and inflammation-blocking drugs have failed (Cavaillon et al, 2020). More recent studies have emphasised the importance of metabolic reprogramming in sepsis progression (Cavaillon and Skirecki, 2019). Sepsis induces a biphasic metabolic response, beginning with an acute hypermetabolic phase characterised by robust inflammation, enhanced catabolic activity, and suppression of anabolic pathways, which may subsequently transition to a hypometabolic state (Wasyluk and Zwolak, 2021). During this early phase, immune activation, enhanced phagocytosis, acute-phase protein production, and systemic physiological responses such as tachycardia, tachypnoea and fever impose markedly higher energy demands. Despite these increased metabolic energy needs, sepsis is commonly accompanied by anorexia (Vandewalle and Libert, 2022), which is observed by a lack of food intake in sepsis mouse models (Vandewalle et al, 2021) and patients (Miller et al, 2021). This mismatch between increased energy demand and limited nutrient intake represents a fundamental metabolic challenge during sepsis. Aviello et al (2021) summarise that loss of appetite during inflammation is a highly conserved behavioural response with protective effects for the host (Aviello et al, 2021). However, increasing experimental evidence suggests that sickness-induced anorexia can have context-dependent consequences and may worsen host outcomes under certain conditions. For example, anorexia appears to be detrimental during viral infection, while being protective during a *Listeria* bacterial infection (Wang et al, 2016). Consistent with this complexity, during *Salmonella Typhimurium* infection, blocking IL-1β-mediated anorexia enhances host survival but simultaneously promotes pathogen transmission (Rao et al, 2017), highlighting a trade-off between host defence and pathogen fitness. Overall, Jindal et al (2024) conclude in their review that the beneficial effects of anorexia depend, at least in part, on the type of pathogen causing the infection and how infection-induced metabolic and tissue stress responses influence host immunity and pathogen fitness (Jindal et al, 2024). Beyond experimental animal models, nutritional support strategies have been investigated in clinical studies. A recent meta-analysis demonstrates that early enteral nutrition, initiated within 24–48 h of intensive care unit (ICU) admission or sepsis diagnosis, significantly reduces all-cause mortality in adult sepsis patients compared with non-early enteral nutrition strategies (Xu et al, 2025). However, the included randomised controlled trials (RCTs)

---

[1]Mouse Genetics in Inflammation, VIB Center for Inflammation Research, Ghent, Belgium. [2]Department of Biomedical Molecular Biology, Ghent University, Ghent, Belgium. ✉E-mail: tineke.vanderhaeghen@irc.vib-ugent.be

**Glossary**

| | |
|---|---|
| Anorexia | A reduction or loss of appetite leading to less food intake, which is commonly observed during infection or inflammation. |
| Blood–brain barrier | A highly selective barrier formed by endothelial cells, pericytes, and astrocytes lining brain blood vessels that protects the central nervous system from toxins and pathogens, but allowing essential nutrients to pass through. |
| Cachexia | A complex metabolic syndrome associated with chronic disease like cancer, characterised by continuous weight loss, muscle wasting, and often fat loss. This alteration cannot be fully reversed by nutritional support alone. |
| Circumventricular organs | Specialised regions located around the ventricles of the brain that lack a complete blood–brain barrier, enabling direct sensing of circulating hormones, nutrients, and inflammatory signals, and communication with the central nervous system. |
| Enteral nutrition | Nutrients are directly delivered into the gastrointestinal tract via oral intake or feeding tube, using liquid formulas to support digestion and prevent malnutrition in patients who cannot eat sufficient solid food, such as critically ill patients. |
| Hypothalamus-pituitary-adrenal (HPA) axis | A neuroendocrine network connecting the hypothalamus, pituitary gland, and adrenal cortex that regulates cortisol (humans) and corticosterone (rodents) release to coordinate stress, infection, and metabolic responses. |
| Pro-inflammatory cytokines | A group of signalling molecules, including interleukins and tumour necrosis factor family members, that are released during immune activation and promote inflammation during illness. |
| Sepsis | A life-threatening condition which is characterised by a dysregulated host response to infection, leading to organ dysfunction and systemic inflammation. |
| Sickness behaviour | An adaptive response to infection or stress, characterised by fever, fatigue, loss of appetite (anorexia), and social withdrawal, mediated by immune-to-brain signalling to conserve energy and promote recovery. |

use different feeding formulas and caloric targets, limiting further subgroup analyses. Accumulating evidence suggests that optimal caloric strategies in sepsis are phase-dependent. During the very early acute phase of sepsis (days 1–2), permissive underfeeding may be more beneficial than high-energy intake (Owais et al, 2014; Sun et al, 2021; Wang et al, 2024). When permissive underfeeding is prolonged (Pardo et al, 2023), it leads to cumulative energy deficits associated with increased mortality, supporting a shift toward greater energy provision during the late acute phase. Consistent with this, a recent retrospective cohort study shows that a higher energy intake during the later acute phase (3–7 days) is not associated with secondary infections in patients with septic shock, while non-shock patients who achieve ≥50% of their energy targets exhibit a significantly reduced incidence of fungal and intra-abdominal infections (Wen et al, 2025). Furthermore, using a deep learning model, Wang et al (2024) have reported that during the late acute phase, appropriate overfeeding may enhance survival by safely compensating for the energy deficit accumulated during earlier permissive underfeeding (Wang et al, 2024).

Altogether, these studies indicate that both the timing and the amount of nutritional intake influence sepsis outcome. Findings from animal models and clinical studies suggest that feeding strategies that are either excessive or insufficient may be harmful, supporting the concept that nutritional support should be adapted to the different phases of disease. However, most current experimental and clinical approaches focus on forced feeding or exogenous energy supplementation, which primarily compensate energy deficits without directly addressing the mechanisms underlying sepsis-associated anorexia. By contrast, appetite-stimulating agents are widely used in the treatment of cancer cachexia (Li and Ling, 2024), yet clinical studies investigating the causes and treatment of anorexia in septic patients are lacking. This gap highlights the need to better define the mechanisms driving anorexia during sepsis and to identify therapeutic targets that could restore appetite or modulate feeding behaviour. Addressing these mechanisms may provide a more physiological strategy for nutritional support and potentially improve metabolic regulation and clinical outcomes in sepsis.

# The hypothalamus, the central coordinator of feeding behaviour

In order to regulate feeding behaviour, the brain is constantly receiving signals from the periphery to maintain energy stores and energy needs. Furthermore, the brain can sense and respond to inflammation present in the periphery, like during sepsis, and adjust feeding behaviour and metabolic processes. Therefore, understanding the central mechanisms the brain is using to regulate feeding behaviour will provide new insights into the metabolic adaptations necessary during inflammation and can lead to the identification of potential therapeutic targets or interventions (Kim et al, 2018; Andermann and Lowell, 2017). Within the brain, the hypothalamus is the key coordinator of various physiological processes such as regulating feeding behaviour, systemic energy homeostasis and thermoregulation (Goel et al, 2025). This requires a complex interplay between several nuclei present in the hypothalamus namely, the arcuate nucleus (ARC), the paraventricular nucleus (PVN), the ventromedial nucleus of the hypothalamus (VMN), the lateral hypothalamic area (LHA), and the dorsomedial hypothalamic nucleus (DMN) (Clarke et al, 2024; Timper and Brüning, 2017) (Fig. 1). Regarding appetite regulation, the ARC is the most extensively studied and well-characterised brain region. It is located at the mediobasal hypothalamus (MBH) near the median eminence (ME), one of the circumventricular organs (CVOs) of the brain. CVOs are rich in fenestrated capillaries leading to a leaky blood–brain barrier (BBB) allowing the passive diffusion of blood-borne molecules, including essential nutrients like glucose and amino acids, and

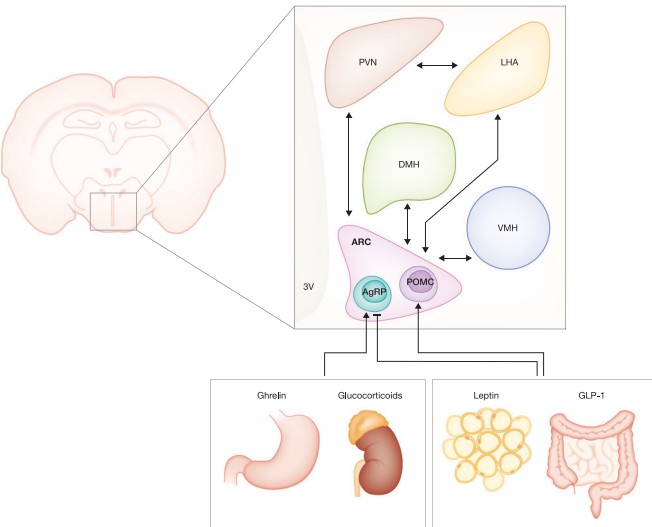

**Figure 1. Schematic representation of the neuronal populations within the hypothalamus regulating homeostatic food intake.**

Key brain sites present in the hypothalamus are involved in appetite regulation, with the ARC being the most extensively studied and well-characterised region, lining the 3V. The ARC contains the orexigenic AgRP neurons and the anorexigenic POMC neurons. These populations integrate hormonal information coming from the periphery (ghrelin, leptin, GLP-1, glucocorticoids) and communicate with other nuclei within the hypothalamus to regulate feeding behaviour. 3V third ventricle, ARC arcuate nucleus, VMH ventromedial nucleus of the hypothalamus, DMH dorsomedial hypothalamus, LHA lateral hypothalamic area, PVN paraventricular nucleus, AgRP agouti-related peptide, POMC pro-opiomelanocortin, GLP-1 glucagon-like peptide 1. Created in BioRender. https://BioRender.com/ai8m09z.

signalling molecules such as hormones and cytokines (Rodríguez et al, 2010) (Fig. 2).

The ARC contains two major and well-studied neuronal populations involved in appetite regulation. Within a conventional view, these populations are distinct and functionally antagonistic, namely the orexigenic agouti-related peptide (AgRP)/neuropeptide Y (NPY)-expressing neurons and the anorexigenic pro-opiomelanocortin (POMC)/cocaine and amphetamine-regulated transcript (CART)-expressing neurons. POMC neurons are activated in response to energy intake and suppress food intake as a consequence. Upon activation, POMC neurons release many neuropeptides, among which alpha-melanocyte-stimulating hormone (α-MSH) is best characterised, after post-translational cleavage of the POMC precursor polypeptide (Timper and Brüning, 2017; Andermann and Lowell, 2017). These POMC-derived peptides gradually promote satiety and increase energy expenditure through binding to and activation of their receptors, namely the melanocortin 3 ($MC_3$) and $MC_4$ receptors, expressed in the PVN and the nucleus of the solitary tract (NTS) in the brainstem (Fenselau et al, 2017). α-MSH is a key component of the hypothalamus in transmitting appetite-suppressing signals, as central administration of α-MSH in mice causes reduced food intake, while mice lacking α-MSH and POMC exhibit hyperphagia and obesity, which can be reverted by α-MSH administration (Wu et al, 2023). Notably, accumulating evidence indicates that POMC neurons are functionally heterogeneous, and their activation does not always oppose feeding. Under specific physiological or experimental conditions, stimulation of POMC neurons can elicit behavioural responses that resemble those induced by AgRP/NPY neurons, including the promotion of food intake (Koch et al, 2015). This apparent functional diversity is likely attributed to molecular

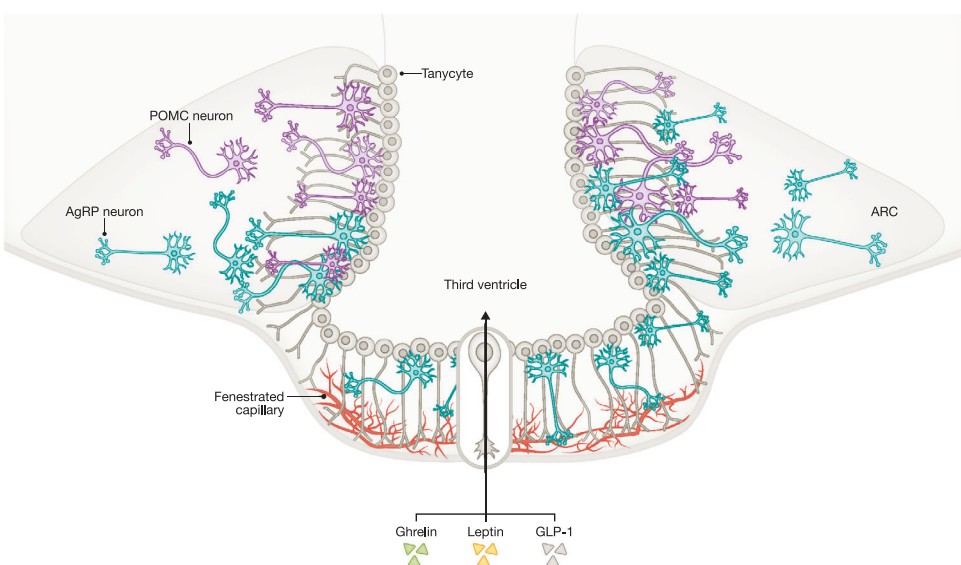

**Figure 2. Circumventricular organs – How tanycytes deliver hormonal signals to the arcuate nucleus.**

The ARC is located at the mediobasal hypothalamus near the median eminence, one of the circumventricular organs (CVOs) in the brain. CVOs are rich in fenestrated capillaries, allowing passive diffusion of blood-borne molecules such as ghrelin, leptin, and GLP-1. Tanycytes, cells lining the third ventricle, can transport hormones from the blood into the third ventricle cerebrospinal fluid, where they can be sensed by the neurons present in the ARC. The ARC contains two functionally distinct antagonistic neuronal populations regulating appetite, namely the AgRP and POMC neurons. ARC arcuate nucleus, AgRP agouti-related peptide, POMC pro-opiomelanocortin, GLP-1 glucagon-like peptide 1. Created in BioRender. https://BioRender.com/66y384c.

and anatomical heterogeneity within the POMC neuronal population. Single-cell RNA sequencing studies have revealed distinct POMC neuron subtypes, including populations expressing low levels of POMC alongside high AgRP expression, as well as canonical POMC neurons with high POMC and minimal AgRP expression, which can be further segregated into four distinct clusters. Quarta et al (2021) comprehensively reviewed the classification strategies used to define POMC neuron subtypes and discussed how this heterogeneity relates to appetite regulation (Quarta et al, 2021). In line with this heterogeneity, Biglari et al (2021) demonstrated that leptin receptor-expressing (Lepr⁺) and glucagon-like peptide-1 receptor-expressing (Glp1r⁺) POMC neurons exhibit minimal anatomical overlap and exert divergent effects on feeding, with selective activation of POMC$^{Lepr^-}$ neurons producing only modest anorexia, whereas stimulation of POMC$^{Glp1r^-}$ neurons robustly promotes satiety (Biglari et al, 2021). Furthermore, lineage tracing combined with single-cell profiling has uncovered a distinct population of POMC-lineage neurons, termed "ghost neurons", that express negligible levels of POMC. The abundance of these cells increases in diet-induced obesity independently of neurogenesis or cell death and can be reversed by weight loss, highlighting the dynamic nature of POMC neuron identity and function (Leon et al, 2024).

In contrast to POMC neurons, AgRP/NPY neurons are activated in response to a negative energy balance and are inhibited by feeding (Andermann & Lowell, 2017). Upon activation, the AgRP peptide is released and promotes food intake by antagonising the $MC_3$ and $MC_4$ receptors and thereby causes hyperphagia when centrally administered (Ollmann et al, 1997; Rossi et al, 1998) or genetically overexpressed in mice (Graham et al, 1997). In addition to AgRP, these neurons also secrete NPY and γ-aminobutyric acid (GABA). NPY directly stimulates food intake via activation of the NPY 1 (NPY1R) and 5 (NPY5R) receptors located in the PVN, DMN, and LHA (Clarke et al, 2024). Furthermore, NPY is required for the long-lasting effects of AgRP neurons on feeding behaviour (Chen et al, 2019). Finally, GABA is the most abundant and fast-acting neurotransmitter in the ARC. Upon release by AgRP neurons, it inhibits PVN satiety neurons and thereby controls feeding behaviour (Tong et al, 2008) (Fig. 3). Recently, a study performed by De Solis et al, (2024) showed that simultaneous activation and inhibition of AgRP/NPY and POMC neurons, respectively, is necessary to promote food intake (De Solis et al, 2024), which challenges the classical Yin and Yang control of feeding behaviour by the AgRP/NPY and POMC neurons.

## Ghrelin: the hunger hormone

Ghrelin was discovered in 1999 as the endogenous ligand of the growth hormone secretagogue receptor type 1a (GHSR1a). It was extracted from rat stomach (Kojima et al, 1999) and it is the only known orexigenic peptide hormone produced in peripheral organs (Tschöp et al, 2000; Nakazato et al, 2001). Ghrelin mRNA is mainly expressed by the X/A-like cells in the gastric fundus (Date et al, 2000), but it is also expressed to a lesser extent in the kidneys, intestine, pancreas and placenta. It encodes preproghrelin, a precursor peptide of 117 amino acids, which can be post-translationally processed into at least five products, of which acyl-ghrelin (AG) and unacylated ghrelin (UAG) are the most biologically relevant (Yanagi et al, 2018). Ghrelin-o-acyltransferase

(GOAT) is responsible for the acylation of UAG into AG (Yang et al, 2008), which is required for ghrelin to specifically bind and activate its receptor GHSR1a. Obestatin is also translated from the ghrelin mRNA and is known to antagonise the appetite-stimulating effect of ghrelin. However, the identity of the endogenous receptor of obestatin remains controversial (Zhang et al, 2005). Tanycytes, cells lining the third ventricle, can transport ghrelin from the blood into the cerebrospinal fluid (CSF) present in the third ventricle or can carry ghrelin from the third ventricle to the parenchymal area where it can be sensed by the ARC (Collden et al, 2015; Gomez et al, 2024).

The orexigenic effect of ghrelin is mediated by the GHSR1a, which was proven by Ghsr knock-out mice (Sun et al, 2004). GHSR1a is highly expressed by AgRP/NPY neurons and VMH neurons (Tannenbaum et al, 1998; Nogueiras et al, 2004). Recently, Barrile et al (2023) have shown that the GHSR1a is also homogeneously expressed in neurons present in the LHA and local injection of ghrelin into the LHA increases food intake. The orexigenic effect of ghrelin in the LHA requires the indirect recruitment of orexin neurons of the LHA and the activation of the neurons present in the ARC. This indicates that GHSR1a-expressing neurons in the LHA are also part of the neuronal circuit inducing food intake (Barrile et al, 2023).

Within the ARC, but also VMH, ghrelin binds to cellular GHSR1a, causing increases in intracellular $Ca^{2+}$ levels. These elevated $Ca^{2+}$ levels activate calcium/calmodulin-dependent protein kinase kinase 2 (CaMKK2), which consequently leads to the phosphorylation of AMP-activated protein kinase (AMPK)(Anderson et al, 2008; Hawley et al, 2005). Ghrelin also activates the Sirtuin 1 (Sirt1)-p53 pathway, and promotes AMPK phosphorylation (Velásquez et al, 2011). Altogether, this causes the inactivation of acetyl-CoA carboxylase (ACC), leading to reduced levels of malonyl-CoA, more carnitine palmitoyltransferase A1 (CPT1A) activity, and increased fatty acid (FA) metabolism as a consequence (López et al, 2008). Finally, also CPT1C-mediated ceramide metabolism plays a role in ghrelin-induced feeding. These metabolic changes potentiate the glutamate release from the VMH neurons onto the AgRP/NPY neurons (Ramírez et al, 2013). Ghrelin also binds to GHSR1a and stimulates mTORC1-pS6K1 signalling. Together with the increased expression of the transcription factors pCREB, FOXO1, and BSX1, AgRP and NPY mRNA levels are increased, resulting in food intake (Martins et al, 2012; Stevanovic et al, 2013). Next to CaMKK2, ghrelin also stimulates calcium/calmodulin-dependent protein kinase 1D (CaMK1D)-mediated transcriptional control of orexigenic neuropeptides in AgRP neurons (Fig. 4). Deletion of CaMK1D in AgRP but not POMC neurons is sufficient to abrogate the appetite-stimulating effect of ghrelin. Lack of CaMK1D attenuates CREB phosphorylation and CREB-dependent expression of AgRP/NPY, and reduces the amount of AgRP fibre projections to the PVN (Vivot et al, 2023).

## Hypothalamic regulation of satiety

### Leptin

Peripheral nutrients and hormones regulate the body's energy balance through the central nervous system (CNS). Especially, leptin has been extensively studied and is essential in suppressing appetite and improving energy expenditure (Xu et al, 2011; Liu

## SECOND ORDER NEURONS

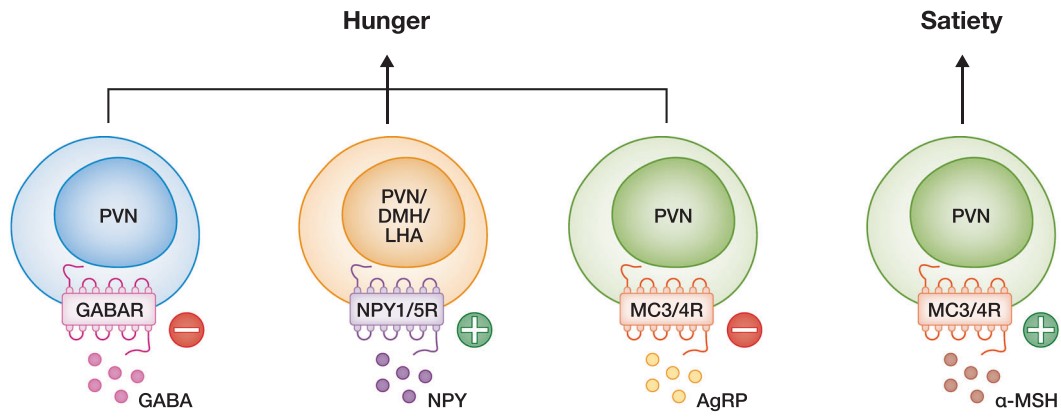

## ARCUATE NUCLEUS

## SENDING SIGNALS

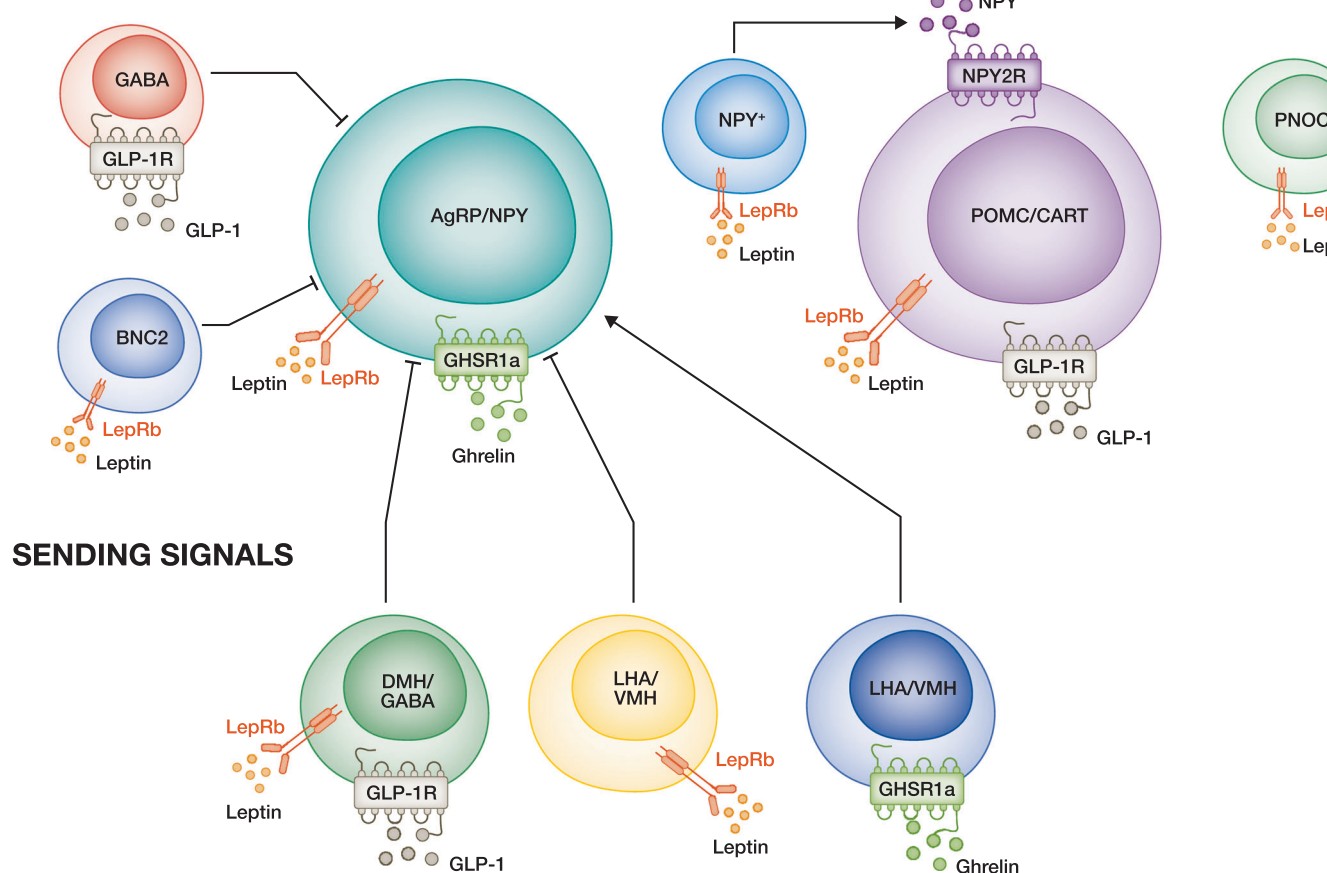

et al, 2023b). While the white adipose tissue (WAT) is the main source of leptin, it is also expressed in other tissues such as the bone marrow, the lymphoid organs, the placenta, the ovaries, and the mammary epithelium (Mantzoros et al, 2011; Boyle et al, 2025). Leptin exerts its physiological functions by binding to leptin receptors (LepRs), which are expressed by cells in various tissues, especially in the hypothalamus (ARC, VMH, DMH, LHA). At least 6 isoforms, LepRa–LepRf, have been identified in rodents, with

LepRb being the primary signalling receptor necessary for the central effects of leptin and mediating most of its known physiological functions (Lee et al, 1996). Leptin (Ob/Ob) or LepRb (Db/Db) deficient mice, as well as humans with analogous deficiencies, exhibit pronounced hyperphagia and obesity, underscoring a crucial role for leptin signalling in regulating both food intake and energy balance (Zhang et al, 1994; Cohen et al, 2001). Furthermore, intracerebroventricular (ICV) injection of leptin into

**Figure 3.  Hormonal signals are integrated at the level of the arcuate nucleus to regulate appetite.**

Sending signals: In response to hunger, ghrelin is directly sensed via GHSR1a expressed on the AgRP/NPY neurons. Furthermore, ghrelin can also bind to GHSR1a on neurons in the LHA and VMH, which causes appetite-stimulating projections towards AgRP/NPY neurons. Leptin and GLP-1 are satiety hormones that are sensed by different neuronal populations. First, POMC/CART neurons are directly activated by the binding of leptin and GLP-1 to their receptors LepRb and GLP-1R, respectively. Furthermore, leptin can also bind to the newly identified PNOC and BNC2 neurons. AgRP-negative NPY neurons are also stimulated by leptin and release NPY that activates POMC neurons by binding to NPY2R. Leptin also reduces AgRP/NPY neuronal activity by binding to LepRb. Additionally, leptin affects LepRb-expressing neurons in the LHA, VMH and DMH, which causes appetite-suppressing projections towards the arcuate nucleus (ARC). Finally, GLP-1 directly activates POMC/CART neurons and indirectly represses AgRP/NPY neurons via the DMH GABAergic neurons by binding to GLP-1R. Also, direct binding of GLP-1 to its receptor expressed on GABAergic neurons present in the ARC projects inhibitory signals towards AgRP/NPY neurons. Regulation of second-order neurons: Upon activation of POMC neurons, these neurons release α-MSH, which binds to the $MC_3$ and $MC_4$ receptors expressed on the neurons present in the PVN. When ghrelin is detected by AgRP/NPY neurons, these neurons are activated and release AgRP, NPY and GABA. AgRP antagonizes $MC_3$ and $MC_4$ receptors. NPY directly stimulates neuropeptide $Y_1$ and $Y_5$ receptors expressed on the neurons located in the PVN, DMH and LHA. GABA is the most abundant and fast-acting neurotransmitter in the ARC. When GABA is released from AgRP neurons, it inhibits PVN satiety neurons by binding to its receptor GABAR. Altogether, these satiety and hunger signals regulate feeding behaviour. AgRP agouti-related peptide, NPY neuropeptide Y, POMC pro-opiomelanocortin, CART cocaine and amphetamine-regulated transcript, PNOC prepronociceptin, BNC2 basonuclin 2, GABA γ-aminobutyric acid, α-MSH alpha-melanocyte stimulating hormone, VMH ventromedial nucleus of the hypothalamus, LHA lateral hypothalamic area, DMH dorsomedial hypothalamus, PVN paraventricular nucleus, GABAR γ-aminobutyric acid receptor, NPY1/2/5R neuropeptide $Y_1/Y_2/Y_5$ receptor, MC3/4R melanocortin 3/4 receptor, GHSR1a growth hormone secretagogue receptor type 1a, LepRb leptin receptor isoform b, GLP-1 glucagon-like peptide 1, GLP-1R glucagon-like peptide 1 receptor. Created in BioRender. https://BioRender.com/5299sv3.

Ob/Ob mice or restoring LepRb signalling in the CNS of Db/Db mice completely reversed the obesity phenotype, highlighting the importance of the CNS in exerting the effects of leptin (de Luca et al, 2005).

As LepRb lacks intrinsic kinase activity, it requires the non-receptor tyrosine kinase Janus kinase 2 (JAK2) for signal transduction. Upon leptin binding, LepRb homodimerizes, leading to the autophosphorylation and activation of JAK2. Subsequently, JAK2 phosphorylates tyrosine residues (Tyr985, 1077 and 1138) of LepRb, thereby causing the recruitment and phosphorylation of signal transducers and activators of transcription 3 (STAT3) (Kisseleva et al, 2002). Upon phosphorylation, STAT3 homodimerizes which enables STAT3 translocation to the nucleus and transcriptional initiation of target genes including suppressor of cytokine signalling 3 (SOCS3), an inhibitor of the LepRb-JAK2 signalling pathway (Fig. 4). Furthermore, also STAT5 and protein tyrosine phosphatase non-receptor type 11 (PTPN11, also known as SHP2) are recruited and activated by JAK2 contributing to the downstream effects of LepRb (Hekerman et al, 2005; Gong et al, 2007). Remarkably, recruitment of SOCS3 to Tyr985 suppresses LepRb signalling (Bjorbak et al, 2000). Next to the JAK2/STAT3 signalling pathway, which is the most important pathway for exerting leptin's actions, other important signalling pathways of leptin include the PI3K/Akt pathway, the ERK1/2 pathway, and the AMPK signalling. How leptin is signalling through these pathways is recently reviewed by Hristov (2025).

LepRb is nicely distributed within the brain, particularly in hypothalamic nuclei responsible for appetite regulation and energy homeostasis, namely the ARC, VMH, DMH and LHA (Liu et al, 2023b). Within the ARC, leptin acts on two major neuronal populations. It directly stimulates POMC neurons, which promotes the release of α-MSH and CART, and suppresses food intake. In parallel, leptin inhibits AgRP/NPY neurons by binding to its receptor (Myers et al, 2008; Coppari and Bjørbæk, 2012). More recent studies have revealed a more complex leptin signalling in the ARC. AgRP neurons have been classically defined by NPY co-expression, however ~20% of NPY neurons in the ARC do not express AgRP. Lee et al (2020) have shown that LepRb is more expressed in this subpopulation (Lee et al, 2020). Stimulation of these NPY$^+$LepRb$^+$ neurons affects energy expenditure and a delayed increase in food intake under basal conditions (Lee et al, 2023). Furthermore, stimulation of these AgRP-negative NPY neurons causes an elevation in food intake, which is even more pronounced within an energy surplus condition. NPY originating from AgRP-negative NPY neurons activates POMC neurons by binding to NPY2R, which plays a central role in the leptin responsiveness of these POMC neurons (Qi et al, 2023). Also, novel ARC populations which are responsive to leptin have been identified, such as prepronociceptin (PNOC)(Solheim et al, 2025) and basonuclin 2 (BNC2)(Tan et al, 2024) expressing neurons. Loss of LepRb expression in PNOC-expressing neurons causes hyperphagia and weight gain in mice, while restoring LepRb expression in PNOC neurons on a *Lepr*-null background reduces body weight. Feeding behaviour is affected as PNOC neurons are able to regulate NPY expression (Solheim et al, 2025). BNC2 neurons directly inhibit AgRP neurons, thereby suppressing food intake (Fig. 3). These neurons are responsive to leptin, as deletion of LepRb in BNC2 neurons causes hyperphagia, similarly to what is detected in AgRP-specific LepRb knock-out mice, identifying BNC2 as a crucial component in appetite regulation (Tan et al, 2024).

Several secreted, intracellular factors have been identified as modulators of leptin signalling in POMC neurons. First, the expression of the angiopoietin-like growth factor (AGF) in POMC neurons is increased by leptin-induced STAT3 phosphorylation upon refeeding after overnight starvation. This suggests that AGF might act as a downstream mediator of leptin in POMC neurons (Jang et al, 2021). Second, a recently identified neuropeptide is spexin (SPX). ICV administration of SPX significantly increases POMC mRNA levels, associated with reduced food intake. Leptin promotes SPX mRNA expression via STAT3 signalling in LepRb-expressing POMC neurons. This suggests that SPX is involved in how leptin regulates feeding behaviour through POMC transcription (Jeong et al, 2022). A final modulator is growth factor receptor-bound protein 10 (GRB10), an adaptor protein that interacts with the insulin and leptin receptors. Ablation of GRB10 in AgRP neurons promotes weight gain, while its overexpression in the AgRP neurons reduces body weight. Furthermore, its deletion or overexpression in POMC neurons induces obesity or reduces food intake and body weight, respectively. GRB10 enhances the

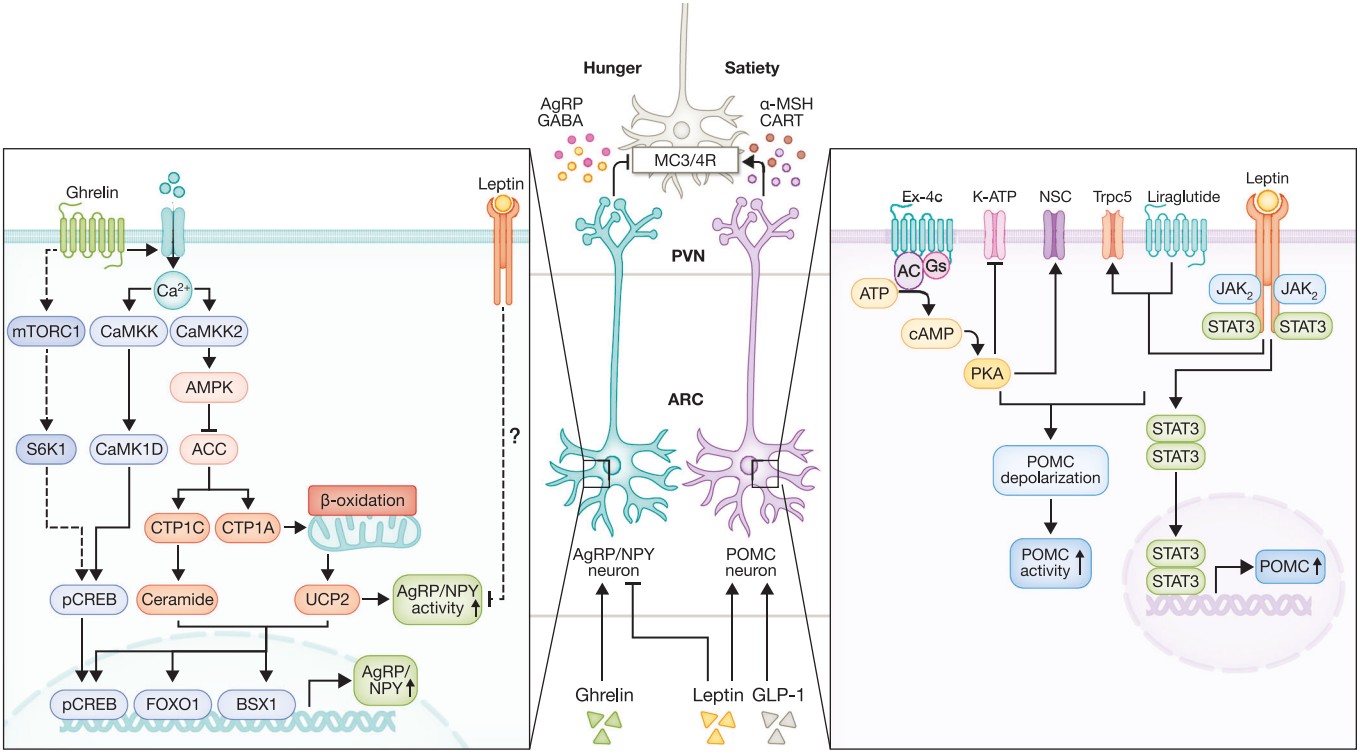

**Figure 4. AgRP/POMC signalling in regulating feeding behaviour.**

AgRP/NPY neurons: Ghrelin binds to GHSR1a, leading to an increase in intracellular $Ca^{2+}$ and activation of multiple signalling pathways. One possible pathway involves $Ca^{2+} \rightarrow$ CaMKK2 $\rightarrow$ AMPK, which inhibits ACC and increases CPT1 activity. CPT1 has two relevant isoforms: CPT1A, which facilitates the transport of long-chain fatty acids into mitochondria for β-oxidation, promoting ROS production and subsequently activating UCP2, and CPT1C, which enhances ceramide signalling. These signalling events increase the activity of the transcription factors pCREB, FOXO1 and BSX1, leading to elevated AgRP and NPY mRNA expression and ultimately stimulating food intake. $Ca^{2+}$ can also signal via CaMKK $\rightarrow$ CaMK1D $\rightarrow$ pCREB, further enhancing AgRP/NPY transcription. Additionally, ghrelin activates mTORC1 $\rightarrow$ S6K1 $\rightarrow$ pCREB, reinforcing AgRP/NPY upregulation. These pathways collectively increase AgRP/NPY neuronal activity and promote feeding. Leptin suppress AgRP/NPY neuronal activation. Activated AgRP neurons release AgRP and GABA, which enhance hunger. POMC neurons: Leptin binds to LepRb and activates JAK2 $\rightarrow$ STAT3, increasing POMC mRNA expression. The GLP-1RA liraglutide activates POMC neurons via TrpC5 channels, while Ex-4c signals through the AC $\rightarrow$ cAMP $\rightarrow$ PKA pathway. This signalling closes $K_{ATP}$ channels, depolarises POMC neurons. Activated POMC neurons release α-MSH and CART, which act on $MC_3/MC_4$ receptors to induce satiety. AgRP agouti-related peptide, NPY neuropeptide Y, GHSR1a growth hormone secretagogue receptor type 1a, $Ca^{2+}$ calcium ion, CaMKK2 calcium/calmodulin-dependent protein kinase kinase 2, AMPK AMP-activated protein kinase, ACC acetyl-CoA carboxylase, CPT1 carnitine palmitoyltransferase 1, ROS reactive oxygen species, UCP2 uncoupling protein 2, pCREB phosphorylated cAMP response element-binding protein, FOXO1 forkhead box protein O1, BSX1 brain-specific homeobox protein 1, CaMK1D calcium/calmodulin-dependent protein kinase 1D, mTORC1 mechanistic target of rapamycin complex 1, S6K1 ribosomal protein S6 kinase beta-1, POMC pro-opiomelanocortin, LepRb leptin receptor long isoform b, JAK2 janus kinase 2, STAT3 signal transducer and activator of transcription 3, TrpC5 transient receptor potential canonical channel 5, GLP-1RA glucagon-like peptide-1 receptor agonist, Ex-4c exendin-4(1-32)-K-capric acid, AC adenylyl cyclase, cAMP cyclic adenosine monophosphate, PKA protein kinase A – Arrows indicate activation, and perpendicular bars indicate inhibition. Created in BioRender. https://BioRender.com/cyw56s4.

inhibitory effect of leptin on AgRP neurons, while it stimulates the excitatory effect of leptin on POMC neurons, mainly through regulating ion channel activity (Liu et al, 2023a).

### GLP-1

Glucagon-like peptide (GLP-1) is a small peptide hormone, derived by the post-translation cleavage of the preproglucagon peptide, encoded by the glucagon gene *Gcg*. It is mainly secreted by the intestinal L-cells, specialised enteroendocrine cells present in the ileum and colon, and by preproglucagon neurons present in the NTS. Secretion of GLP-1 is regulated by several factors involved in homeostasis (Müller et al, 2019; Hwang et al, 2025). GLP-1 is released at low levels during fasting, whereas its secretion significantly increases after eating (Campbell and Drucker, 2013). Besides nutrient-driven secretion, leptin also stimulates the release

of GLP-1 (Bodnaruc et al, 2016). To exert its physiological effects, GLP-1 binds to its receptor (GLP-1R). Preclinical studies have identified central GLP-1Rs as key mediators in the regulation of feeding and energy balance by GLP-1. ICV injection of GLP-1 suppresses food intake in wild-type mice, while this response is completely abolished in whole body GLP-1R knock-out mice (Scrocchi et al, 1996). Furthermore, blocking central GLP-1R by administering exendin-9, a GLP-1R antagonist, drastically increases food intake and disrupts glucose regulation without affecting body weight (Knauf et al, 2008). Also, CNS-specific deletion of GLP-1R in mice leads to an increased food intake without affecting body weight (Sisley et al, 2014). Altogether, GLP-1 in the brain has a significant role in energy and glucose metabolism. As endogenous GLP-1 is rapidly degraded by dipeptidyl-peptidase 4, resulting in a short half-life, long-acting GLP-1 analogues or GLP-1R agonists

(GLP-1RA) have been developed and are able to target multiple nuclei within the brain, including the hypothalamus (Gabery et al, 2020).

The hypothalamus, and more specifically the ARC, has emerged as a target for both peripherally administered GLP-1RAs and NTS-derived GLP-1 in order to regulate food intake and glucose homeostasis (Hwang et al, 2025). To enter the ARC, GLP-1R-expressing tanycytes are involved in the transport of GLP-1RAs across the BBB, as deletion of GLP-1Rs in tanycytes abolishes the entrance of liraglutide, a GLP-1 analogue, into the hypothalamus and inhibits the weight loss effect as a consequence (Imbernon et al, 2022). Moreover, fluorescently labelled GLP-1RAs accumulate within the ARC upon systemic administration, which is completely absent in GLP-1R knock-out mice (Gabery et al, 2020), highlighting the requirement of GLP-1 binding to its receptor and the necessity of GLP-1R signalling in the ARC to achieve weight loss. POMC neurons express GLP-1Rs, and liraglutide is able to directly activate these neurons, in which transient receptor potential channel 5 (TrpC5) channels are activated (Secher et al, 2014; He et al, 2019). Furthermore, liraglutide indirectly increases the excitatory tone of POMC neurons (Péterfi et al, 2021), which contributes to the weight-reducing effect of liraglutide (Fig. 4).

In contrast to POMC neurons, AgRP/NPY neurons are indirectly inhibited by GLP-1RAs. More specifically, GLP-1R-expressing GABAergic neurons located in the ARC and DMH are activated upon GLP-1RAs administration and provide an inhibitory input towards AgRP/NPY neurons (Kim et al, 2024; Webster et al, 2024). Furthermore, thyrotropin-releasing hormone (Trh[+]) inhibitory neurons also express GLP-1R and are activated by liraglutide. Activating these Trh[+] neurons, present within the ARC, inhibits AgRP neurons and feeding, while silencing these neurons significantly increases food intake and attenuates the effect of liraglutide (Webster et al, 2024). Altogether, local and distal GLP-1R-expressing GABAergic neurons suppress AgRP/NPY neuronal activity, which mediates the effects of GLP-1RAs such as liraglutide on feeding behaviour.

Next to the ARC, GLP-1R-expressing neurons present in the DMH also play an important role in controlling feeding (Kim et al, 2024). Silencing of GLP-1R-expressing neurons in the DMH abolishes the inhibitory effect of GLP-1RAs on AgRP/NPY neurons, which causes hyperphagia and obesity (Maejima et al, 2021). On the other hand, chemogenetic activation of these GLP-1R-expressing neurons within the DMH inhibits AgRP/NPY neurons and reduces food intake (Kim et al, 2024). Furthermore, a typical GABAergic neuronal inhibitory input to AgRP/NPY neurons originates from LepRb-expressing neurons present in the DMH. Recently, Rupp et al (2023) showed that some of these neurons coexpress both LepRb and GLP-1R, which are activated by both local and systemic administration of liraglutide. Ablating LepRb from these neurons provoked hyperphagia, while reactivation of the LepRb in GABAergic neurons in LepR null mice required LepRb expression in these GABAergic GLP-1R expressing neurons. Moreover, restoring GLP-1R expression in these LepRb/GLP-1R co-expressing neurons in GLP-1R null mice suppresses food intake by liraglutide (Rupp et al, 2023). Overall, these GABAergic neurons expressing both LepRb and GLP-1R play an important role in regulating food intake (Fig. 3).

# Cytokine and hormonal dynamics during sepsis

Sepsis is characterised by a significant increase in the release of pro-inflammatory cytokines, such as tumour necrosis factor alpha (TNFα), interleukin (IL) 1β and IL-6, detected in the blood of sepsis patients (Molano Franco et al, 2019; Gharamti et al, 2022) and septic mice (Ramsay et al, 2025), which is correlated with disease prognosis. Notably, TNFα and IL-6 levels rapidly rise after sepsis onset and remain high during the chronic phase of sepsis (Carmichael et al, 2023), whereas IL-1β peaks later (Sun et al, 2025). In order to execute a proper response against the inflammation induced by sepsis and to fulfil the required energy needs, the immune system interacts with the CNS and the endocrine system (Mehdi et al, 2025).

Under basal, physiological conditions, the neurons present in the PVN produce corticotropin-releasing hormone (CRH), which stimulates the release of adrenocorticotropic hormone (ACTH) by the corticotrope cells of the pituitary gland. In turn, ACTH induces the production of glucocorticoids (GCs, corticosterone in mice and cortisol in humans) by the adrenal glands. Upon release of GCs into the bloodstream, a GC-mediated negative feedback system inhibits the secretion of CRH and ACTH (Wasyluk and Zwolak, 2021). However, during sepsis, pro-inflammatory cytokines such as TNFα and IL-6 stimulate the release of CRH and ACTH by the hypothalamus and pituitary, respectively, by crossing the BBB. This causes a strong increase in baseline GC levels in the blood (Vandewalle et al, 2021; Annane et al, 2017). Moreover, cytokines can directly stimulate GC synthesis independently of ACTH, resulting in a sustained adrenal GC production during sepsis (Kanczkowski et al, 2015). Upon sepsis progression, the GC secretion and feedback mechanisms of the hypothalamus-pituitary-adrenal (HPA) axis can be disturbed, thereby causing critical illness-related corticosteroid insufficiency (Annane et al, 2017; Téblick et al, 2022). Therefore, GC levels can be very high in response to acute inflammation, or very low when the HPA axis fails. How these stress hormones regulate hunger and satiety during environmental, psychological and especially septic stress remains an outstanding research area.

Next to GCs, anorexigenic factors also significantly alter upon sepsis progression. Leptin acutely increases after sepsis onset and gradually decreases over time, both in sepsis patients (Karampela et al, 2021) and during caecal ligation and puncture (CLP)-induced polymicrobial sepsis (Shapiro et al, 2010). Indeed, leptin was originally considered an IL-6-like cytokine. The main driver responsible for enhanced leptin expression in WAT and release in the blood is inflammation, as administration of TNFα, IL-1β or IL-6 promptly increases serum leptin levels and leptin mRNA levels in WAT (Sarraf et al, 1997). The other major satiety factor, GLP-1, also rises during the early phase of sepsis, which serves as a predictor of clinical outcome. Elevated GLP-1 levels are detected within 24 h after sepsis onset, which remain high until 28 days, and are associated with persistent organ dysfunction and progression to chronic critical illness in sepsis patients. Consistently, plasma GLP-1 levels are significantly higher during lipopolysaccharide (LPS)-induced endotoxemia and CLP-induced polymicrobial sepsis (Lebrun et al, 2023), mediated by a cascade of primarily secreted inflammatory cytokines like IL-1β and IL-6 (Kahles et al, 2014). An

**Box 1 Mouse models for sepsis**

To gain a better insight into the complex cellular and molecular mechanisms underlying sepsis, model organisms are often used, as studying these processes in humans is often not feasible. Mice are commonly used in a preclinical sepsis model. They are widely accessible, easy to breed, and have a well-characterised genome in which mutations can be easily introduced. Several experimental sepsis models have been developed in mice, each with their limitations. These can be classified into three categories, which include (i) the administration of an endogenous toxin such as LPS from Gram-negative bacteria, (ii) inoculation with a viable pathogen, and (iii) the disruption of a protective barrier, thereby allowing bacterial invasion (Cai et al, 2023).

First, to induce a robust, acute, systemic inflammatory response, a single bolus injection of an exogenous toxin such as LPS can be administered. To ameliorate the limitations of a single injection, a continuous infusion is also used to induce severe inflammation. The advantage of this approach is the ease of use and reproducible responses. However, this method does not account for Gram-positive bacteria and polymicrobial sepsis (Copeland et al, 2005). Second, sepsis can also be induced by the exogenous administration of a viable pathogen. Inoculation of mice with live bacteria such as *Escherichia coli*, *Streptococcus pneumoniae*, *Staphylococcus aureus*, etc. is technically easy and reproduces septic shock. However, the immunological response may vary depending on the bacterial strain, and the host response can be altered dependent on the site of infection (Cai et al, 2023). Another approach is the implantation of a specific amount of bacteria in a fibrin clot into the abdominal cavity. This allows the slow release of bacteria and is used to study the acute and chronic phases of sepsis (Ghanta et al, 2021). Finally, three commonly used models of intraperitoneal sepsis are caecal slurry (CS), caecal ligation and puncture (CLP), and colon ascendens stent peritonitis (CASP). In the CS model, caecal content is intraperitoneally injected, resulting in a stronger but shorter acute inflammatory response compared to CLP and CASP. This model is preferably used in neonatal mice, given their small size and the technical ease to induce sepsis (Starr et al, 2014). The CLP model is a regularly used, well-accepted sepsis model, and it is still considered as the golden standard in sepsis research due to its high similarities with human sepsis. Here, the caecal barrier is perforated, thereby causing a peritoneal infection with faecal content. It produces an immune, metabolic, hemodynamic and biochemical response similar to humans. However, it is difficult to control the caecal content release and the intra- and interexperimental variability in severity and survival (Dejager et al, 2011). To induce CASP, a small stent is inserted into the ascending colon, allowing a continuous flow of faecal content into the abdominal cavity. This causes systemic bacteremia, organ infection by bacteria and a profound inflammatory response. The sepsis severity can be adjusted by the size of the stent. Compared to CLP, this model is technically more challenging and has a less pronounced hemodynamic response (Lustig et al, 2007).

Since sepsis is characterised by a profound shift from a hyperinflammatory phase towards immunosuppression, sepsis patients are more susceptible to secondary infections. Therefore, the two-hit model is often used to investigate the immunosuppressive phase in more detail. As a first hit, routine sepsis models are used, such as pathogen inoculation or CLP surgery, to establish an intrinsic infection. During the second hit, mice are frequently injected with a clinically relevant pathogen to mimic the secondary infection that human sepsis patients encounter. Although it mimics the immunosuppressive phase during sepsis, the health status after the first hit is often different, and there is a lack of standardised protocol for the secondary infection. This may further increase inconsistencies in experimental outcome (Muenzer et al, 2006; Wang et al, 2019).

overview of commonly used sepsis mouse models can be found in Box 1.

Next to changes in the blood level of satiety hormones, the levels of the hunger hormone ghrelin also alter upon inflammation. Administration of LPS to healthy volunteers causes an early increase in ghrelin levels (Vila et al, 2007). Furthermore, critically ill patients and patients with intra-abdominal sepsis also exhibit a significant rise in their ghrelin levels (Maruna et al, 2005; Koch et al, 2010). Interestingly, higher ghrelin levels are frequently detected in the blood of adult sepsis patients, and are associated with a shorter ICU stay, a reduced need for mechanical ventilation, and are positively correlated with survival (Nikitopoulou et al, 2020).

In general, these cytokines and hormones are components of the neuroendocrine network that links peripheral organ activity to clinical outcome. Furthermore, they act as well-established regulators of appetite, synthesised by peripheral organs and capable of transmitting signals, either rapidly or continuously, to the CNS via diverse pathways.

# From peripheral signals to central integration during sepsis

Sickness behaviour is a physiological reaction to an acute systemic inflammatory response, often induced by an infection. One of the key aspects that alters, is eating behaviour, characterised by anorexia, weight loss, and thirst. It is considered as an adaptive response to protect patients from local and systemic inflammation. To regulate this sickness behaviour, the brain is activated via two main routes: the neural afferent pathways and humoral signalling. These routes are responsible for transmitting peripheral signals to the brain (Bourhy et al, 2022).

## The neural network

In sepsis, inflammatory mediators released at the site of inflammation can activate peripheral nerves as these express specific receptors for cytokines (such as TNFα, IL-1β and IL-6), damage-associated molecular patterns, and pathogen-associated molecular patterns, including LPS. Nearly all tissues are innervated by a dense network of sensory endings, enabling the peripheral nervous system to detect a broad spectrum of physiological and pathological signals (Gautron and Layé, 2009).

Among the peripheral nerves, the vagal nerve is the most studied one, involved in the immune-brain crosstalk (Fig. 5). It regulates metabolic homeostasis by controlling food intake, gastrointestinal motility and secretion, hepatic glucose production, and other visceral functions. Furthermore, the vagal nerve controls inflammatory responses during pathogen invasion and tissue injury, known as the inflammatory reflex. Indeed, systemic inflammation induces robust sickness behaviour in a vagal nerve-dependent way and generates neural responses in the NTS. Vagal sensory fibres are activated upon close contact with mucosal immune cells or directly via pro-inflammatory cytokines such as TNFα, IL-1β or IL-6. During sepsis, vagotomy significantly increases mortality in LPS-induced endotoxemia and colon ascendens stent peritonitis induced polymicrobial sepsis, whereas vagal nerve stimulation

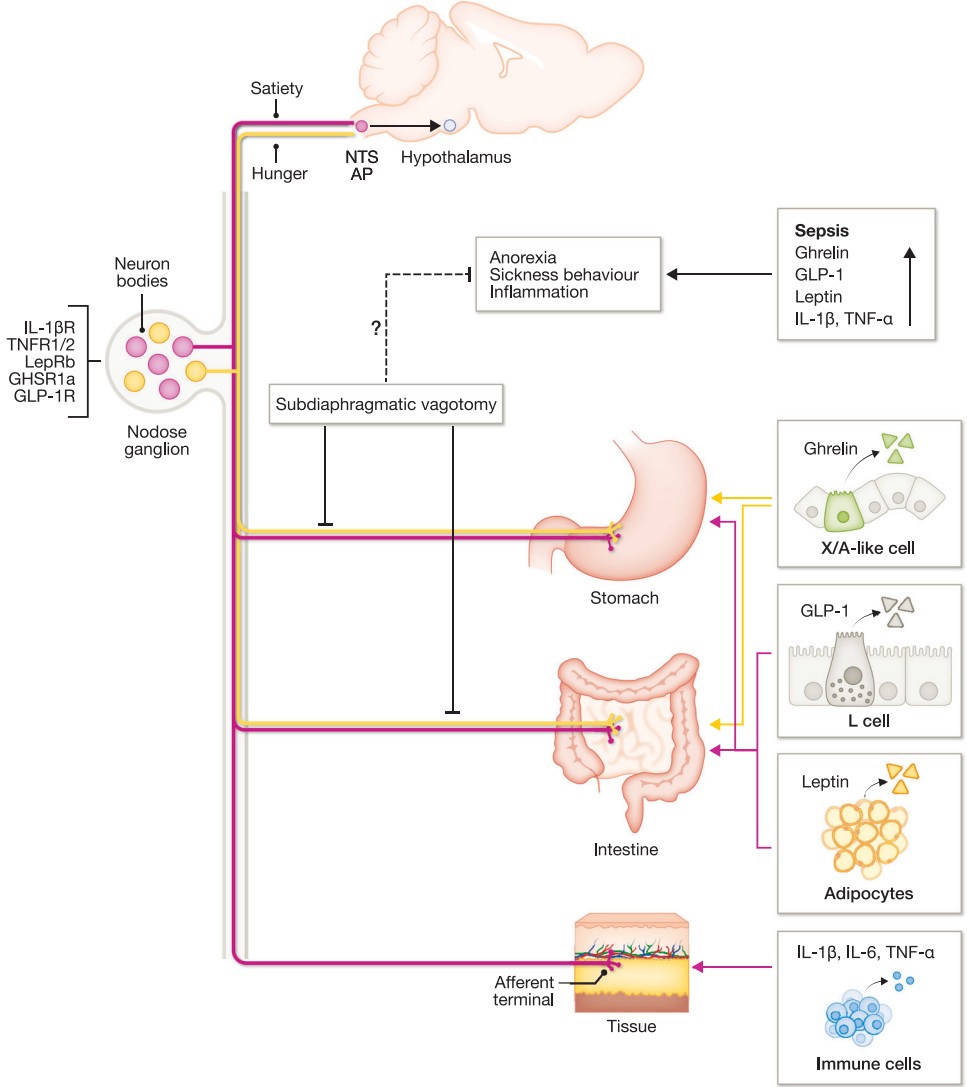

**Figure 5.  The vagal nerve in appetite regulation during homeostasis and inflammation.**

The vagal visceral afferents, of which the cell bodies are present in the nodose ganglia, project to the NTS and AP. Afferent signalling is further communicated via neural contacts within the brain, such as the hypothalamus, to regulate feeding behaviour. Vagal sensory fibres are activated upon close contact with mucosal immune cells or directly via pro-inflammatory cytokines such as TNFα, IL-1β or IL-6. Next to sensing inflammation, stomach and gut-innervating vagal afferent neurons (VANs) are influenced by ghrelin, leptin and GLP-1, and communicate by binding to their receptors on these VANs, thereby affecting vagal nerve-mediated feeding behaviour. Nodose ganglia neurons can also directly respond to pro-inflammatory cytokines and hormones by specifically expressing their receptors. During sepsis, pro-inflammatory cytokines and hunger and satiety hormone levels increase, thereby inducing anorexia. How subdiaphragmatic vagotomy affects anorexia during sepsis remains controversial. TNFα tumour necrosis factor alpha, IL-6 interleukin 6, IL-1β interleukin 1 beta, GLP-1 glucagon-like peptide 1, IL-1βR interleukin 1 beta receptor, TNFR1/2 tumour necrosis factor receptor 1/2, LepRb leptin receptor long isoform b, GHSR1a growth hormone secretagogue receptor type 1a, GLP-1R glucagon-like peptide 1 receptor, NTS nucleus tractus solitarius, AP area postrema. Created in BioRender. https://BioRender.com/ycuehlr.

attenuates the systemic inflammatory response to LPS (Borovikova et al, 2000; Huang et al, 2010; Kessler et al, 2012). Furthermore, Huerta et al, (2025) have shown that nodose ganglia neurons directly respond to TNFα and IL-1β (Huerta et al, 2025), by specifically expressing TNF receptor 1 and 2 or the IL-1β receptor (Steinberg et al, 2016). IL-1β also activates vagal afferents using transient receptor potential ankyrin 1 ion channels (Silverman et al, 2023). Next to sensing inflammation, stomach and gut-innervating vagal afferent neurons (VANs) are also influenced by ghrelin, by communicating through its receptor GHSR1a expressed on these

VANs (Perelló et al, 2022). VAN-specific knockdown of GHSR1a influences feeding behaviour, as it increases meal frequency accompanied by a reduced meal size (Davis et al, 2020). Moreover, within the gastrointestinal tract, GLP-1R is expressed on vagal sensory neurons in the stomach and intestine (Williams et al, 2016). Stimulation of these GLP-1R-expressing vagal afferents rapidly inhibits AgRP neurons, thereby reducing food intake (Bai et al, 2019). Both GHSR1a and GLP-1R co-localise in the nodose ganglia innervating the stomach, where ghrelin and GLP-1 influence each other in terms of vagal nerve-mediated feeding

(Zhang et al, 2020). Finally, also VANs expressing LepRb (Iwasaki et al, 2013) in their nodose ganglia are present within the stomach and duodenum (Fig. 5). Deletion of leptin signalling in these VANs is sufficient to promote hyperphagia (de Lartigue et al, 2014).

These vagal afferents of which the cell bodies are located in the nodose ganglia, transmit the peripheral signals via the synaptic connections to the NTS and area postrema (AP). These brainstem nuclei function as key integrative hubs that convey peripheral information to the hypothalamus, thereby mediating feeding and sickness-related behaviours. The NTS and AP contain distinct neuronal populations that regulate food intake, including those expressing LepRb and GLP-1R. A more detailed discussion of the different neuronal populations and their connectivity with hypothalamic feeding circuits has been comprehensively reviewed by Johansen et al, (2025). In the context of inflammatory and septic conditions, GLP-1 signalling within the caudal brainstem appears to play a critical role in inflammation-induced anorexia. Notably, fourth-ventricular administration of the GLP-1R antagonist exendin-(9–39) significantly attenuates the reduced food intake following systemic LPS treatment, highlighting the contribution of caudal brainstem GLP-1R signalling to LPS-induced anorexia (Grill et al, 2004). Beyond GLP-1-responsive neurons, recent studies have identified specific subsets of neurons in the NTS and AP that are acutely activated following LPS administration, as indicated by Fos expression. Among these, ADCYAP1[+] neurons have been shown to be sufficient to recapitulate LPS-induced sickness behaviour, while inhibition of this neuronal population markedly alleviates anorexia, adipsia, and locomotor suppression (Ilanges et al, 2022). Collectively, these findings identify the NTS and AP as promising targets for the regulation of inflammation-associated anorexia. However, studies in this area remain limited, and further work is required to define the specific neuronal populations involved and to elucidate how brainstem circuits engage hypothalamic pathways to control feeding behaviour.

How vagal nerve signalling contributes to inflammation-induced anorexia has been extensively studied in vagotomised animals. Bilateral subdiaphragmatic vagotomy or chemical deafferentation using capsaicin attenuates inflammation-induced anorexia and brain IL-1β expression upon peripheral administration of LPS or IL-1β (Bret-Dibat et al, 1995; Layé et al, 1995). Furthermore, bilateral vagotomy in rats restores feeding upon exposure to low doses of LPS; food intake cannot be restored upon administration of high LPS doses (Sergeev and Akmaev, 2000). More recently, Bourhy et al, (2024) revealed that subdiaphragmatic vagotomy effectively reduces acute brain activation at the NTS and inflammatory responses, and alleviates sickness behaviour triggered by CLP-induced polymicrobial sepsis (Bourhy et al, 2024). These contradictory outcomes might reflect the dual and context-dependent functions of vagal nerve signalling. The vagal afferents sense peripheral inflammatory mediators and transmit these signals to the CNS, contributing to sickness behaviour and anorexia, while efferent vagal output engages the cholinergic anti-inflammatory reflex to restrain excessive cytokine release and prevent tissue injury in multiple models of systemic inflammation, shock, and sepsis (Huston, 2012). Disruption of vagal signalling may be beneficial in attenuating maladaptive behavioural responses during mild or acute inflammatory challenges, but might be harmful in severe or systemic inflammatory states where vagal nerve-mediated anti-inflammatory control is needed. The effect of vagotomy might depend on the inflammatory context, disease severity, and the balance between afferent and efferent vagal pathways engaged. Further studies are needed to investigate how the effects of

vagotomy depend on the state of inflammation and disease severity to clarify the context-dependent role of vagal signalling across septic conditions.

## Humoral pathways

While neural signalling is rapid, humoral signalling takes longer to establish but exerts long-lasting effects. Circulating metabolic and inflammatory factors can reach the hypothalamus via two major pathways, namely the BBB and CVOs (Bourhy et al, 2022). In essence, the BBB is a layer of endothelial cells that separates the CNS from systemic circulation (Obermeier et al, 2013). In response to inflammatory mediators (e.g. TNFα, IL-6 and IL-1β produced during sepsis), tight junctions are disrupted, thereby enhancing BBB permeability and allowing cytokines to penetrate the brain (Peng et al, 2021). Once these inflammatory mediators enter the brain, the brain-resident immune cells, such as microglia and astrocytes, get activated (van Gool et al, 2010; Barichello et al, 2023). Prolonged microglial activation amplifies the release of pro-inflammatory cytokines and reactive oxygen species (ROS), further disrupting the BBB and promoting neuronal death (Akrout et al, 2009).

Next to cytokines, also hormones enter the brain across the BBB (Fig. 6). Leptin is transported into the brain via an active, saturable mechanism involving its receptor LepRb, which is expressed by the endothelial cells of the BBB (Di Spiezio et al, 2018). Furthermore, tanycytes lining the MBH activate LepRb upon peripheral leptin administration, thereby facilitating the passage of leptin from circulation into the CSF and enabling leptin-induced STAT3 activation in the ARC, VMH and DMH hypothalamic nuclei to regulate food intake (Balland et al, 2014; Duquenne et al, 2021). On the contrary, Yoo et al, (2019, 2020) have demonstrated that selective deletion of LepR in tanycytes does not affect leptin-induced pSTAT3 expression in hypothalamic neurons. Furthermore, using mice in which tanycytes in the ARC and ME are conditionally ablated, they have observed that tanycytes are not required for the active transport of leptin from the blood into the hypothalamus (Yoo et al, 2019, 2020). Although central administration of leptin is required for an improved immune response during CLP-induced polymicrobial sepsis (Tschöp et al, 2010), LPS significantly downregulates leptin transport across the BBB in a dose-dependent manner and with a long-lasting effect (Nonaka et al, 2004). Ghrelin is transported across the BBB in a GHSR1a-independent manner (Rhea et al, 2018). Furthermore, the uptake of ghrelin into the brain via tanycytes occurs in a GHSR1a-independent and clathrin-dependent manner (Gomez et al, 2024). Also, passive diffusion of ghrelin across fenestrated capillaries in the ME occurs (Schaeffer et al, 2013). Although total and acyl-ghrelin levels increase during sepsis (Nikitopoulou et al, 2020), further investigation is necessary to identify whether ghrelin uptake into the brain, either via the BBB or via tanycytes, is altered upon systemic inflammation.

# Sickness signals in the arcuate nucleus: rewiring appetite during sepsis

AgRP neurons are activated by a negative energy balance and directly respond to hunger signals like ghrelin and low blood glucose. Conversely, these neurons are less active in the presence of

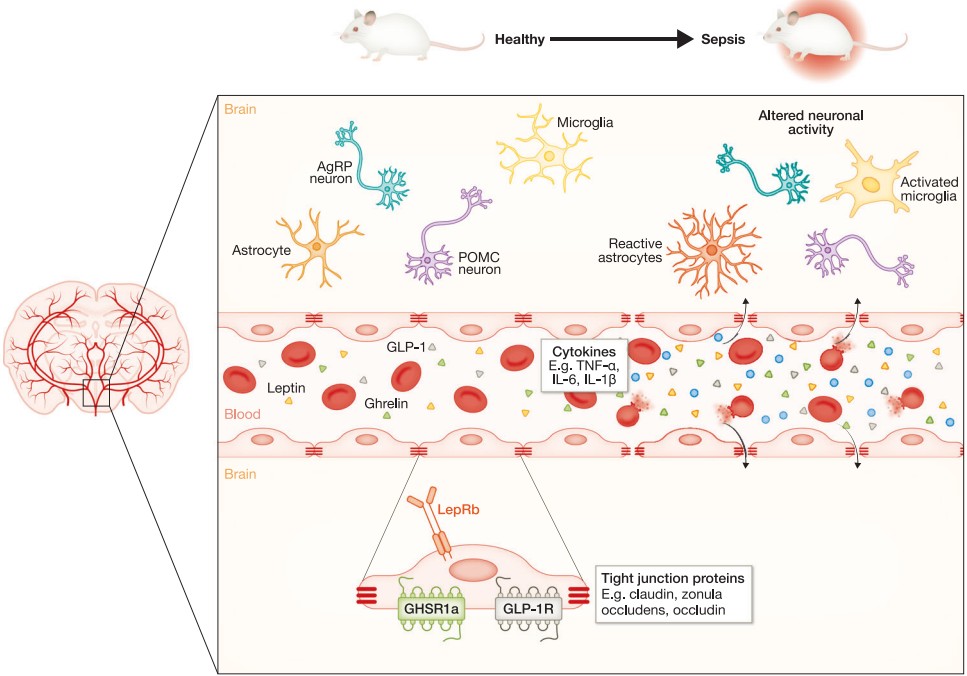

**Figure 6.   From peripheral inflammation to hypothalamic injury: the blood–brain barrier in sepsis.**

The blood–brain barrier (BBB) is a layer of endothelial cells (ECs) separating the central nervous system from systemic circulation. ECs express several tight junction proteins and hormone receptors (GHSR1a, LepRb, GLP-1R), thereby allowing active transport of ghrelin, leptin and GLP-1 into the brain. This is sensed by AgRP and POMC neurons, but also by astrocytes and microglia, which communicate with the neuronal populations in order to regulate food intake. During sepsis, increased levels of cytokines such as TNFα, IL-6 and IL-1β, and hormones like ghrelin, leptin and GLP-1 are present in the blood. As tight junction connections between ECs are lost, the blood-borne molecules can passively diffuse into the brain. Astrocytes and microglia are activated by the pro-inflammatory cytokines, and also the neuronal function of AgRP and POMC neurons is altered during sepsis, which affects their responses towards the increased hormone levels and their capacity to regulate feeding behaviour. AgRP agouti-related peptide, POMC pro-opiomelanocortin, GHSR1a growth hormone secretagogue receptor type 1a, LepRb leptin receptor isoform b, GLP-1 glucagon-like peptide 1, GLP-1R glucagon-like peptide 1 receptor, TNFα tumour necrosis factor alpha, IL-6 interleukin 6, IL-1β interleukin 1 beta. Created in BioRender. https://BioRender.com/67s5lve.

high-energy levels and are inhibited by anorexigenic factors, such as leptin and GLP-1 (Andermann and Lowell, 2017). It is well-established that LPS-induced endotoxemia induces anorexia (Holubová et al, 2018). However, how inflammation affects AgRP neurons and suppresses appetite remains unclear (Aviello et al, 2021). Hao et al, (2016) have assessed AgRP neuron activation using cFos expression and have shown that LPS-induced endotoxemia blunts fasting-induced cFos activation and refeeding responses, suggesting impaired engagement of canonical hunger-related transcriptional programs (Hao et al, 2016). In contrast, Su et al, (2017) and Boutagouga Boudjadja et al, (2022) have directly measured neuronal activity using electrophysiological or calcium-based approaches and reported that AgRP neuron activity can be sustained in fasted mice during endotoxemia (Su et al, 2017; Boutagouga et al, 2022). These observations suggest that inflammatory states may uncouple neuronal firing from immediate early gene induction and behavioural output. Moreover, Osterhout et al, (2022) have recently shown that AgRP neurons can be directly suppressed by neurons present in the ventromedial preoptic area, which are activated upon LPS injection (Osterhout et al, 2022). Importantly, despite preserved AgRP neuronal activity, multiple studies have shown that optogenetic or chemogenetic activation of AgRP neurons fails to restore food intake following LPS administration (Liu et al, 2016; Essner et al, 2017), indicating that

inflammatory signals impose circuit-level constraints downstream of AgRP neurons. Together, these findings support a model in which AgRP neuron activity is maintained or altered in inflammatory contexts but is insufficient to drive feeding behaviour, likely due to upstream inhibitory inputs and/or downstream override of feeding circuits.

Next to the AgRP neurons, the ARC also contains the satiety-promoting POMC neurons. As mentioned above, these neurons are activated in response to energy intake, thereby suppressing food intake (Aviello et al, 2021). Systemic administration of LPS significantly decreases food consumption and is associated with increased levels of POMC and CART mRNA in the ARC (Sergeyev et al, 2001). On the other hand, other studies did not detect changes in POMC mRNA expression or neuronal activity upon LPS challenge (Borges et al, 2007). However, hypothalamic injection of LPS also causes a significant reduction in food intake, which is associated with increased hypothalamic pro-inflammatory cytokine production, and is inhibited by blocking hypothalamic nuclear factor-kappa B (NFκB) activity. NFκB also directly stimulates POMC transcriptional activity, implying the involvement of NFκB in controlling feeding behaviour. Indeed, blocking IκB kinase β, an upstream kinase of NFκB, in POMC neurons attenuates LPS-induced anorexia (Jang et al, 2010). Moreover, Quarta et al, (2021) have identified different subtypes of POMC neurons (Quarta et al,

2021). Until now, no research has been published in which has been investigated what the role of these different POMC neuronal subtypes is during systemic inflammation or sepsis. Therefore, we believe it would be of great interest to address how these different subpopulations of POMC neurons behave under septic conditions and whether one of these subtypes can be identified as a potential target to reverse sepsis-associated anorexia.

## Pro-inflammatory cytokines

Upon inflammation, pro-inflammatory cytokines affect the ARC, and more specifically, the AgRP/NPY and POMC neurons, in terms of feeding behaviour and body weight. During systemic inflammation or acute infection, these inflammatory factors cause anorexia and weight loss, while obesity is favoured upon nutrient excess (Thaler et al, 2010). For example, Chaves et al, (2020) demonstrated that TNFα and IL-1β acutely inhibit the activity of AgRP neurons up to 42%, whereas only a few POMC neurons are depolarised by TNFα. Also, IL-6 does not induce any acute change in the activity of both AgRP and POMC neurons, indicating that the activity of AgRP neurons is primarily affected by TNFα and IL-1β (Chaves et al, 2020), which may contribute to inflammation-induced anorexia. Also, ICV administration of TNFα in rats results in a significant reduction of AgRP, NPY, and POMC mRNA expression levels, which is associated with an inactivation of AMPK signalling. Unfortunately, blocking the activity of TNFα by ICV administration of infliximab did not restore food intake in CLP-induced polymicrobial sepsis (Arruda et al, 2010).

Besides TNFα, Scarlett et al, (2008) have shown that ICV administration of IL-1β significantly increases cFos and AgRP mRNA levels in the ARC, mediated by the type I IL-1 receptor (IL-1RI) expressed by these neurons. Although AgRP mRNA expression levels increase by IL-1β, this cytokine inhibits AgRP secretion from hypothalamic explants (Scarlett et al, 2008). Furthermore, POMC neurons also express the IL-1RI, and ICV injection of IL-1β results in a higher cFos activity and action potential of POMC neurons, and stimulates the release of α-MSH (Scarlett et al, 2007). Altogether, IL-1β produces a strong anorexigenic effect by enhancing hypothalamic $MC_4$ receptor signalling, which contributes to inflammation-induced anorexia. On the contrary, neither central nor peripheral administration of IL-6 affects food intake (Wallenius et al, 2002). Interestingly, central administration of both IL-6 and IL-1β, using doses that do not affect food intake alone, induces anorexia (Harden et al, 2008). These findings demonstrate that IL-6 modulates hypothalamic circuits controlling food intake in a context-dependent manner, acting synergistically with other pro-inflammatory cytokines rather than as a primary anorexigenic signal. Altogether, these data indicate that inflammatory mediators affect AgRP/NPY and POMC mRNA expression levels in a post-transcriptional manner, thereby altering their translation, synthesis or release, ultimately reducing food intake.

Moreover, emerging evidence indicates that glial activation can also modulate hypothalamic feeding circuits. LPS, a toll-like receptor 4 (TLR4) agonist expressed on ARC microglia, suppresses orexigenic AgRP/NPY neuronal activity, whereas the activity of the majority of POMC neurons is increased. This results in altered feeding behaviour and supports a role for microglia in sickness-associated anorexia. Importantly, the inhibitory effects of LPS on AgRP/NPY-mediated feeding are abolished by pharmacological inhibition of microglial function or by blockade of TLR4 signalling (Reis et al, 2015). Jin et al, (2016) have shown an important role for TLR2 in stimulating hypothalamic microglia to promote POMC neuronal activation, thereby contributing to sickness behaviour like anorexia, hypoactivity, and hyperthermia. Antagonists of NF-κB, the cyclooxygenase pathway and $MC_3/MC_4$ receptors reverse the body weight loss induced by TLR2 activation (Jin et al, 2016). Varela et al, (2021) indicate that AgRP neurons and neighbouring astrocytes engage in bidirectional interactions. AgRP neurons release GABA to induce mitochondrial and structural adaptations in neighbouring astrocytes, which in turn enhances AgRP neuron excitability via increased ensheathment and astrocyte-derived prostaglandin E2 signalling (Varela et al, 2021). How these interactions are modulated during sepsis remains largely unexplored.

## Appetite-regulating hormones

As mentioned earlier, leptin acutely increases during sepsis, both in patients (Karampela et al, 2021) and during CLP-induced polymicrobial sepsis (Shapiro et al, 2010). In the ARC, AgRP neurons express LepRb and deletion of LepRb from these neurons increases daily food intake and body weight, whereas LepRb deletion from POMC neurons does not affect these parameters. GABAergic afferents on AgRP neurons are necessary to acutely inhibit appetite by leptin. On the contrary, post-synaptic activation of $K_{ATP}$ channels is required for the chronic inhibition of AgRP neurons by leptin (Xu et al, 2018). LPS-induced endotoxemia causes pronounced anorexia in both Ob/Ob and Db/Db mice, in which Db/Db mice are resistant to LPS-induced anorexia, while Ob/Ob mice are more sensitive to loss of appetite induced by LPS. These data indicate that leptin per se might not be essential for LPS-induced anorexia (Faggioni et al, 1997). However, neutralising circulating leptin using leptin antiserum (LAS) during LPS-induced endotoxemia clearly increases food intake and body weight in rats. Whereas LAS does not affect circulating IL-6 levels upon LPS challenge, IL-1β mRNA levels are significantly reduced in the hypothalamus. This indicates that leptin is a circulating mediator of LPS-induced anorexia using a hypothalamic IL-1β-dependent mechanism (Sachot et al, 2004).

Next, also GLP-1 rises rapidly after sepsis onset, which has been detected in sepsis patients, LPS-induced endotoxemia and CLP-induced polymicrobial sepsis (Lebrun et al, 2023). GLP-1 is known to regulate food intake, at least in part through hypothalamic circuits. Evidence suggests that GLP-1 or a GLP-1RA can directly stimulate POMC/CART neurons by binding to the GLP-1R, while indirectly suppressing AgRP/NPY neuron activity via GABA-dependent signalling. This clearly indicates that GLP-1R expressed on POMC/CART neurons mediates the GLP-1-induced weight loss and anorexia (Secher et al, 2014). Furthermore, Yoo et al, (2025) indicate that the GLP-1RA exendin-4(1-32)K-capric acid (Ex-4c) activates POMC neurons using the protein kinase A-dependent signalling pathway. In turn, $K_{ATP}$-sensitive channels are closed, thereby causing depolarisation of the POMC neurons and thus reduced food intake (Yoo et al, 2025). Liraglutide, another GLP-1RA, also enhances POMC neuron excitability via Trp5C channels, while suppressing AgRP/NPY neurons using GABAergic signalling (He et al, 2019). Although the excitability of AgRP/NPY and POMC neurons is altered upon acute inflammation (Chaves et al, 2020) and

GLP-1 levels are elevated during sepsis (Secher et al, 2014), further research is necessary to elucidate how increased GLP-1 levels contribute to sepsis-induced anorexia, which mechanisms GLP-1 is using and whether blocking GLP-1 signalling might alleviate sickness behaviour, and more specifically the reduced food intake, during sepsis.

Several studies implicate that food intake and body weight gain is reduced in the absence of endogenous GC production, induced by surgical adrenalectomy (Cohn et al, 1955; Green et al, 1992). Removal of the adrenal glands increases the sensitivity to leptin (Zakrzewska et al, 1997). Upon stress or septic conditions, baseline GC levels in the blood are increased (Annane et al, 2017; Vandewalle et al, 2021). These elevated GC levels cause the direct activation of AgRP and NPY mRNA transcription, dependent on the GC receptor (GR), which promotes food intake (Lee et al, 2013; Yoshimura et al, 2023). Furthermore, deletion of GR in AgRP/NPY neurons leads to increased energy expenditure and decreased food intake, accompanied by weight loss (Shibata et al, 2016). Unfortunately, how GCs and GR directly regulate AgRP/NPY and POMC expression during sepsis requires further investigation.

Finally, ghrelin expression levels increase rapidly after sepsis onset, and decrease over time during sepsis progression. Upon LPS-induced endotoxemia, hypothalamic ghrelin and GHSR1a mRNA levels are significantly reduced, which is associated with a reduced expression of AgRP and NPY, and an increase in POMC and CART mRNA expression (Duan et al, 2022). Peripheral administration of ghrelin does not promote changes in food intake and body weight, AMPK activity and AgRP/NPY mRNA expression in LPS-induced endotoxemia (Rivas et al, 2017). On the contrary, long-acting, stable GHSR1a analogues significantly increase food intake when rats are challenged with LPS via the activation of orexigenic pathways (Holubová et al, 2018). Also, both peripheral and central administration of ghrelin during CLP-induced polymicrobial sepsis increases hypothalamic GHSR1a and AgRP mRNA expression and lowers JAK2/STAT3 signalling. Despite the fact that this was not associated with changes in feeding behaviour, ghrelin could ameliorate sepsis-induced muscle wasting (Duan et al, 2025). Of note, albeit research on the therapeutic effect of ghrelin on improving cancer-related cachexia is well-documented, clinical research investigating the therapeutic potential of ghrelin on sepsis-associated anorexia is required. Although clinical studies have shown higher ghrelin levels associated with improved survival (Nikitopoulou et al, 2020), the interplay between ghrelin and appetite during sepsis demands further investigation. Also, more research is necessary to identify whether the failure of AgRP to induce feeding is due to a loss of function or whether the priority of AgRP neurons has shifted towards mechanisms reducing peripheral inflammation and metabolism.

## Beyond appetite: regulation of systemic inflammation and metabolism by the arcuate nucleus

While most studies have focused on how peripheral inflammatory and hormonal signals influence hypothalamic feeding circuits during sepsis, emerging evidence suggests that these circuits also exert active feedback control over systemic immune and metabolic responses. Recent work has demonstrated that sustained activation of AgRP neurons in fed mice, achieved through chemogenetic approaches, reduces acute TNFα release during LPS-induced endotoxemia (Boutagouga et al, 2022). In addition, AgRP neurons have been demonstrated to be both sufficient and necessary to reduce circulating Ly6C[Hi] classical monocytes during fasting, an effect that occurs independently of actual nutrient availability (Cavalcanti de Albuquerque et al, 2025). Additionally, during short-term fasting, AgRP neurons release NPY and activate downstream NPY1R circuits to stimulate PVH[CRH] neurons, triggering the HPA axis and GC release, which exerts top-down control of hepatic autophagy and rewiring of metabolism, providing adaptation to the negative energy state (Chen et al, 2023). Finally, Klima et al (2023) have uncovered the ability of a central circuit to reduce inflammation upon food deprivation. More specifically, caloric restriction reduces oedema, body temperature and TNFα release in a mouse model of injury-induced peripheral inflammation. These effects could be recapitulated by activating AgRP neurons. These neurons project towards the PVN and robustly reduce inflammation via vagal efferent signalling (Klima et al, 2023). Together, these findings highlight a previously underappreciated role of feeding-related hypothalamic neurons in the regulation of immune function and metabolism.

Beyond immune and metabolic modulation, hypothalamic feeding circuits have been implicated in regulating gut microbiota composition, as activation or inhibition of AgRP or POMC neurons rapidly reshapes microbial profile (Toledo et al, 2025). Given the emerging role of the gut microbiome in sepsis, hypothalamic circuits might influence host responses indirectly via microbiota-dependent mechanisms. For example, Keskey et al (2025) show that gut microbial metabolites, together with pathogen factors, critically affect survival in mouse models of systemic infection relevant to human sepsis (Keskey et al, 2025). Whether coordinated interactions between hypothalamic feeding circuits and the gut microbiome contribute to disease progression during sepsis remains an important open question.

Together, these findings suggest that alterations in AgRP and POMC neuronal activity during sepsis may not only drive anorexia but also modulate peripheral inflammation, host defence and metabolic homeostasis through brain-to-periphery signalling pathways. This bidirectional communication highlights central feeding circuits as potential therapeutic targets to simultaneously influence nutritional behaviour and immune responses in sepsis.

## General conclusion and future perspectives

In general, AgRP and POMC neurons are the central regulators of feeding behaviour. We propose that, during sepsis, simultaneous inhibition of AgRP neurons and activation of POMC neurons might be responsible for sepsis-associated anorexia (Fig. 7). During sepsis, an important question remains unanswered: can ghrelin, the most potent orexigenic hormone, effectively access the brain, either by crossing the BBB or entering the brain via CVOs, by binding to its receptor GHSR1a and exert its physiological effects? Addressing this question is essential in understanding whether ghrelin signalling is impaired during sepsis, which could have important therapeutic applications in regulating appetite in critically ill sepsis patients.

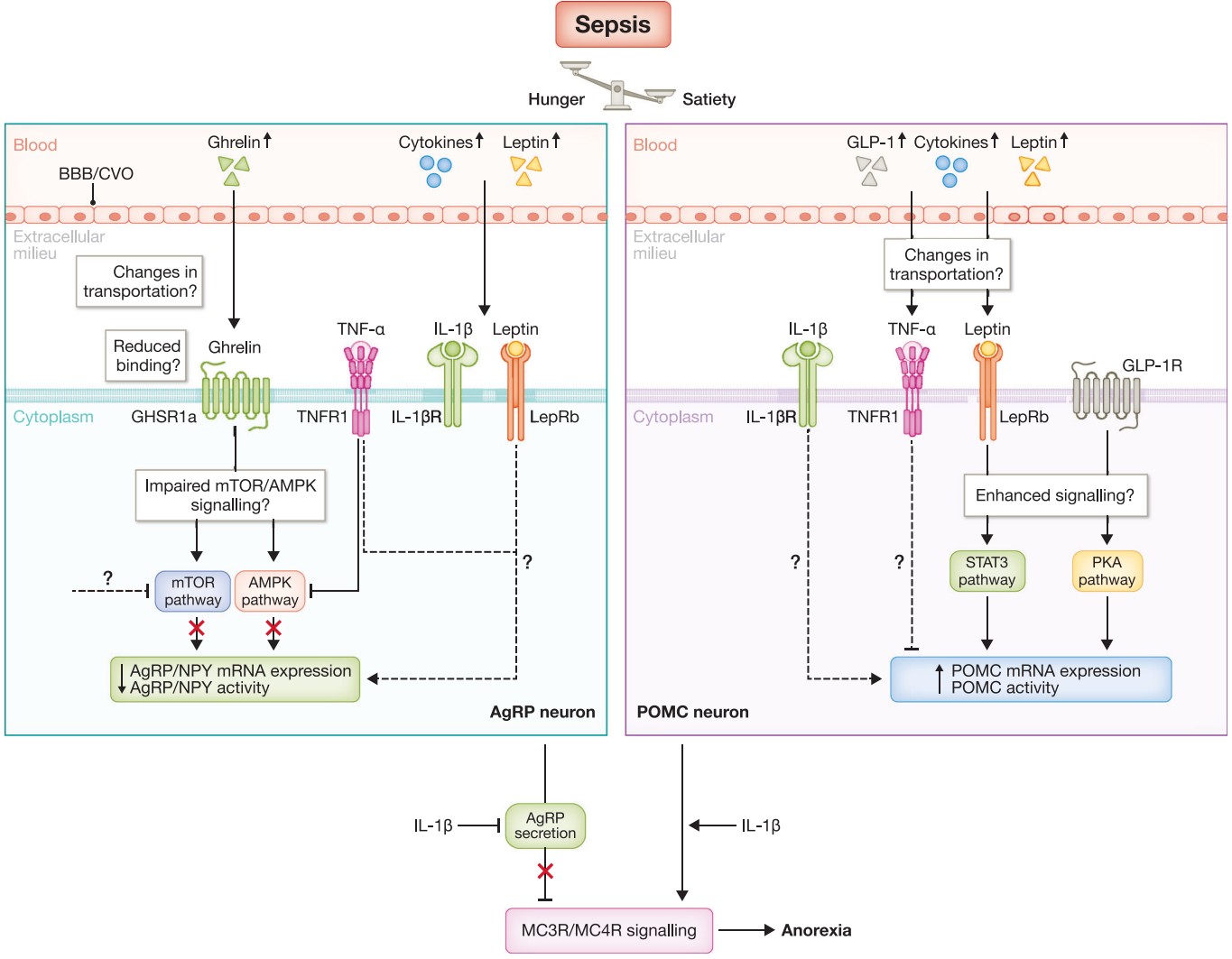

**Figure 7. Potential mechanisms underlying anorexia in the AgRP/POMC neurons during sepsis.**

Anorexia is a hallmark of sepsis and is characterised by reduced activity of orexigenic AgRP neurons and increased activity of anorexigenic POMC neurons. During the acute phase, circulating levels of orexigenic and anorexigenic hormones, as well as cytokines, are elevated. Disruption of the BBB and the permissive nature of CVOs might allow the entrance of these factors into the brain, although their transport efficiency during sepsis remains unclear. Ghrelin, a key hunger hormone, shows a diminished ability to activate AgRP neurons during sepsis, potentially due to a reduced GHSR1a binding or impaired downstream mTOR/AMPK signalling. In contrast, satiety cytokines such as TNFα inhibit AMPK signalling or directly suppress AgRP neuronal activity, thereby promoting anorexia. Leptin activates POMC neurons via STAT3 signalling, inhibits AgRP neuronal activity, and enhances GLP-1-mediated satiety through PKA-dependent pathways. Although IL-1β can increase AgRP transcription, it simultaneously reduces AgRP peptide release and enhances POMC activity, thereby strengthening MC3/MC4 receptor signalling and reinforcing anorexia during sepsis. Overall, these mechanisms might drive sepsis-associated anorexia, highlighting the therapeutic potential of enhancing orexigenic pathways while suppressing anorexigenic signalling to restore feeding during sepsis. AgRP agouti-related peptide, POMC pro-opiomelanocortin, BBB blood–brain barrier, CVOs circumventricular organs, GHSR1a growth hormone secretagogue receptor, mTOR mechanistic target of rapamycin, AMPK AMP-activated protein kinase, TNFα tumour necrosis factor-alpha, STAT3 signal transducer and activator of transcription 3, GLP-1 glucagon-like peptide-1, PKA protein kinase A, IL-1β interleukin 1 beta, MC3R melanocortin 3 receptor, MC4R melanocortin 4 receptor. Created in BioRender. https://BioRender.com/c8fx9ux.

Existing evidence indicates that pro-inflammatory cytokines such as TNFα, IL-1β and IL-6 can suppress AMPK signalling, which is a crucial downstream mediator of ghrelin. Dysregulation of these metabolic pathways might impair AgRP neuron activation. Simultaneously, elevated levels of anorexigenic factors in sepsis, combined with increased permeability of the BBB and CVOs allows the sensing of these satiety factors by the AgRP and POMC neurons. However, it requires further investigation what the precise concentration of these factors within the CSF is during sepsis and how they cross the BBB or enter the ARC via the CVOs. Leptin appears to play a dual role: it suppresses AgRP neuron activity while enhancing the anorexigenic effects of POMC neurons. Furthermore, it amplifies the GLP-1-mediated activation of POMC neurons. Also, research suggests that IL-1β inhibits AgRP peptide secretion and increases POMC mRNA expression, thereby strengthening MC4 receptor signalling and promoting anorexia. Altogether, multiple pathways might drive sepsis-associated anorexia, however current research has not established which

circulating factor(s) might play a dominant role in the ARC. These gaps highlight the need for future research in order to identify the most promising targets to mitigate sepsis-associated anorexia.

Potential therapeutic strategies include both the supplementation of orexigenic factors and the inhibition of anorexigenic signals. One approach is supplementation with ghrelin, which has been shown to effectively improve tumour-associated anorexia and cachexia. However, there is currently no clinical evidence supporting the therapeutic potential of ghrelin in septic patients. A second strategy represents the inhibition of anorexigenic factors: (1) Reducing leptin levels has been shown to increase food intake in mouse models, although its relevance in sepsis remains to be clarified. (2) Blocking TNFα signalling using infliximab has not alleviated anorexia in sepsis mouse models, however research in this area remains limited. Because TNFα blocks AMPK signalling, which might impair ghrelin signalling, combining TNFα inhibitors with ghrelin agonists might have the potential to restore feeding during sepsis. (3) Since GLP-1 levels rise during sepsis, it might be of interest to use GLP-1R antagonists. Currently, there is no evidence that these antagonists are able to restore feeding under septic conditions. (4) Finally, $MC_4$ receptor antagonists have shown efficacy in reversing LPS-induced anorexia and may therefore be a promising therapeutic intervention.

Beyond the AgRP and POMC neurons, multiple hypothalamic neuronal populations and additional anorexigenic signals likely contribute to sepsis-associated anorexia. Therefore, comprehensive mechanistic studies are required to define the underlying mechanisms and guide the development of effective therapeutic interventions.

## Pending issues

1. High quality RCTs are needed to define the optimal timing, amount and composition of nutrition in septic patients and to enable personalised nutrition strategies.
2. How do hunger and satiety factors during sepsis reach the brain and how these signals are integrated within hypothalamic circuits to drive anorexia remains unclear.
3. The key satiety factors and neuronal populations responsible for sepsis-induced anorexia remain unclear.
4. Clinical studies are needed to investigate appetite-targeted therapies, such as ghrelin agonists or GLP-1 receptor antagonists, in septic conditions.

## Peer review information

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

## Acknowledgements

Research in the authors' laboratory was supported by the following grants: Research Project grants from Fonds Wetenschappelijk Onderzoek Vlaanderen 3G014921 and 3G028020, Strategic Basic Research grant by Fonds Wetenschappelijk Onderzoek Vlaanderen 3S003122 and Ghent University grant Methusalem 01M00121. Wanting Zhu holds a China Scholarship Council (CSC) fellowship 01SC0622.

## Author contributions

**Wanting Zhu**: Writing—original draft; Writing—review and editing. **Claude Libert**: Writing—review and editing. **Tineke Vanderhaeghen**: Writing—original draft; Writing—review and editing.

## Disclosure and competing interests statement

Prof. Claude Libert is a member of the EMBO Molecular Medicine Editorial Board. This has no bearing on the editorial consideration of this article for publication. The remaining authors declare no competing interests.

