## [Peer Review File · EMBO Molecular Medicine]

Hypothalamic regulation of sepsis-associated anorexia: cytokine and hormonal signaling through AgRP/POMC circuits

Wanting Zhu, Claude Libert, and Tineke Vanderhaeghen

Corresponding author: Tineke Vanderhaeghen (tineke.vanderhaeghen@irc.vib-ugent.be)

Review Timeline:

Submission Date:	18th Nov 25
Editorial Decision:	10th Dec 25
Revision Received:	14th Jan 26
Accepted:	20th Feb 26

Editor: Lise Roth

Transaction Report:

10th Dec 2025

Dear Dr. Vanderhaeghen,

Thank you for the submission of your review to EMBO Molecular Medicine following your presubmission inquiry. We have now received feedback from the experts who agreed to evaluate your manuscript. As you will see from the reports below, they found the review to be well-written and interesting overall. However, they do raise a few concerns and make suggestions to improve the interest and impact of your work.

We would therefore welcome a revised version of your manuscript that would address these points. Please attach a covering letter giving details of the way in which you have handled each of the points raised by the referees.

- 1/ A .doc formatted version of the manuscript text (including Figure legends and tables).
- 2/ Separate figure files.
- 3/ A letter INCLUDING the reviewer's reports and your detailed responses to their comments.
- 4/ A glossary: EMBO Molecular Medicine articles are accompanied by a glossary explaining some of the terms used for laymen.
- 5/ Pending issues: At the end of each article, there is a box highlighting issues that still need further studies and where research efforts should converge.
- 6/ Please remove the abbreviations list and incorporate the abbreviations in the manuscript text.

For the figures, please note the following points:

- If there are certain aspects of your figure draft that are based upon assumptions or where the scientific data remains ambiguous, please add a comment so that we can work with you on an accurate depiction.
- If the figure or single panels of the figure have been adapted from a published figure, please add this information to the figure legend (e.g., 'Adapted from...' or 'Based on...').
- Please only re-use figures or parts of a figure if this is essential for understanding the concept communicated. If the figure contains re-used images or elements of images, please make sure that you have the permission/license to publish it (this also applies to your own previous work, if the journal you published in retains copyright). All re-used material must be explicitly cited.
- If you use an image data base for scientific iconography (e.g., BioRender), please let us know if you have a license that allows for publication in an academic journal.

Looking forward to receiving your revised manuscript at your earliest convenience,

With kind regards,

Lise Roth

***** Reviewer's comments *****

Referee #1 (Remarks for Author):

In this review, Zhu colleagues address important and timely questions concerning the hypothalamic mechanisms of sepsis-induced anorexia. Although this topic was recently covered in another significant review (PMID: 39326837), the authors' decision to focus on cytokine and hormonal signalling in POMC and AGRP circuits as potential causes of inflammation-induced anorexia is intriguing. This approach offers new mechanistic insights and discussion in this important field. While the topic is interesting, I feel there is room for improvement, particularly in terms of providing clarity and further explanation of several important aspects, as indicated below.

- Chapter 2 provides an in-depth analysis of the role of certain anorexigenic and orexigenic hormones in controlling AGRP and POMC neurons. However, the discussion is not up to date. For example, they overlook significant recent literature on the

heterogeneity of these neurons, particularly with regard to POMC neurons. See, for example, the following key papers: PMCID 34002087, 33633406, 38658557 and 28462073, to name a few. This literature needs to be further integrated. This is particularly relevant given the current evidence suggesting that different POMC neuronal subpopulations may not respond to peripheral metabolic and hormonal signals at all. In this context, it would also be interesting to expand the discussion on how the heterogeneity of these neurons might relate to differential control by inflammatory peripheral signals in the context of sepsis, as discussed in Section 5.

- The review offers a 'neurocentric view' of how inflammatory, hormonal or neuronal signals might impact AGRP or POMC neurons in the context of sepsis. However, it should be noted that sepsis and brain inflammation are also influenced by microglia, astrocytes and peripheral immune cells that can travel to the brain and perpetuate the inflammatory response. Considering the pivotal role of brain immune cells and neuroglia in POMC and AGRP neuronal biology, it is surprising that the authors did not address this issue. I believe this concept warrants integration and further discussion in the manuscript.

- It has recently been shown that hypothalamic circuits in mice can rapidly modulate gut microbiota (PMID: 40263603), which have been postulated to play a role in sepsis (see, for example, PMID: 39779878 or others). It may be interesting to expand on this potential link.

- It has been demonstrated that hypothalamic AgRP neurons exert top-down control over systemic TNF- α release during endotoxemia (PMID: 36182699). AGRP neurons are also both sufficient and necessary for reducing circulating Ly6H classical monocytes during fasting (PMID: 40184437). Therefore, while the authors focus on describing peripheral-to-brain axes that potentially control neural activity during sepsis, they should also consider the importance of brain-to-periphery feedback signals that regulate inflammatory responses during sepsis. Further discussion on this topic is warranted.

- Overall, the manuscript would benefit from a more in-depth analysis of how the authors can critically explain some of the contradictory findings in this field. For example, section 4.1 cites studies showing that vagotomy significantly increases mortality and LPS-induced endotoxemia, while others report the opposite, as the same procedure can apparently reduce inflammation-induced anorexia, inflammatory responses and sickness behaviour. Similarly, in Section 5, the authors report studies in which LPS prevents fasting-induced cFOS activation of AGRP neurons, alongside others showing that, in fasted mice with LPS-induced endotoxemia, AGRP neuron activity is sustained. How can these opposing lines of evidence be reconciled? Can the authors share their thoughts on these contrasting findings?

-In Section 4.2, the authors emphasise the evidence implicating tanycyte cells in the selective uptake of hormones into the hypothalamus. While the cited papers support this hypothesis, others that were not reported do not (PMCID: PMC7423758; PMID: 30941008). Therefore, this concept should be toned down.

-section 5.2. 'It is clear that GLP-1 regulates food intake, as GLP-1 or a GLP-1RA are taken up by POMC/CART neurons by GLP-1R, which directly stimulates these neurons and indirectly inhibits AgRP/PP neurons via GABA-dependent signalling'. Please rephrase this sentence, as it is unclear in its current form.

Referee #2 (Remarks for Author):

This is an interesting and relatively comprehensive review of potential mediators of sepsis-associated anorexia, focusing on the hypothalamus. It is generally well-written and contains several figures, most of which are informative.

Overall, while it is understood that this review focuses on potential hypothalamic mechanisms, there has been a great deal of recent progress in the understanding of roles of the area postrema and nucleus of the solitary tract in this process. It would seem useful to mention these results, if only to place them in context of the hypothalamic circuits.

Some of the subsections contain a great deal of information, but little synthesis. It could be useful to include concluding sentences at the end of each section. While this is true of many subsections, sections 4.1 and 4.2 represent examples of this. Figure 3 is very busy and confusing. One potential way to simplify it could be to note that the Bnc2, Trh, DMH Glp1r, and Lepr/Glp1r neurons all represent the same population of neurons that mediate the inhibition of Agrp neurons. It could also be useful to note the putative functions of the second order neurons.

Minor:

Page 5- THR, CRH, and OXT neurons of the PVH have been shown to play little role in the suppression of food intake.

Last sentence of the top paragraph on page 7- "completely reversed" might make more sense than "completely restored"

Page 8- the discussion of GRB10 seems out of place in a paragraph that otherwise focuses on AGF/SPX. Perhaps it would make more sense to discuss GRB10 in the context of SHP2 (to which GRB10 binds) on page 7.

Referee #3 (Remarks for Author):

The review is timely and is potentially of broad interest. It does a nice job of summarizing many of the key anorexigenic

pathways. However, in its current form it is lacking in several areas. Several suggestions are below.

The entire crux of the article seems to be that better understanding of anorexia during sepsis may lead to potential therapies to treat sepsis-associated anorexia. The entire review seems to be focused on the hypothesis that sepsis induced anorexia is detrimental and should be reversed. However, they do not provide any evidence to support this hypothesis. They mention in the introduction that critically ill patients do not benefit from early full feeding but then never expand on this.

Moreover, the authors could make a better case for why anorexia during sepsis should even warrant treatment. Perhaps an entire section discussing the pros and cons of sepsis-associated anorexia. They could then describe complications such as muscle catabolism, immune suppression, etc. to support their potential treatment strategies.

The article does not explicitly discuss the anti-inflammatory actions of alpha-MSH. It may be worth adding in discussion of how an anti-inflammatory peptide is linked to anorexia. They discuss alpha-MSH only as a "satiety" peptide. Similarly, a discussion on the potential anti-inflammatory effects of ghrelin seems missing.

The authors could include a discussion of central GLP1 signaling inhibits TLR4 signaling. Given the broad use of GLP-1 drugs, this seems relevant.

The authors should also recent work by the Dulac group regarding a preoptic neuronal population controlling fever and appetite during sickness.

Although this review is focused on hypothalamus, the Friedman lab identified a key set of brainstem neurons control multiple aspects of sickness behavior.

Page 3, the sentence "Furthermore, long-term anorexia during sepsis will lead to insufficient energy intake, which is correlated with poor patient outcomes (Kitayama et al 2023). This sentence does not properly reflect this study, which examined the presence of poor appetite 12 months after discharge from ICU (only 10% of admissions were for sepsis), and presence of malignancy at time of ICU admission seems to be driving the vast majority of persistent poor appetite 12 months later.

The authors comment on studies of sickness associated anorexia as an important characteristic of both disease resistance and tolerance during sepsis but don't discuss the work from the Medzhitov group.

REVIEWER 1

In this review, Zhu colleagues address important and timely questions concerning the hypothalamic mechanisms of sepsis-induced anorexia. Although this topic was recently covered in another significant review (PMID: 39326837), the authors' decision to focus on cytokine and hormonal signalling in POMC and AGRP circuits as potential causes of inflammation-induced anorexia is intriguing. This approach offers new mechanistic insights and discussion in this important field. While the topic is interesting, I feel there is room for improvement, particularly in terms of providing clarity and further explanation of several important aspects, as indicated below.

We would like to thank the reviewer for these kind words.

- Chapter 2 provides an in-depth analysis of the role of certain anorexigenic and orexigenic hormones in controlling AGRP and POMC neurons. However, the discussion is not up to date. For example, they overlook significant recent literature on the heterogeneity of these neurons, particularly with regard to POMC neurons. See, for example, the following key papers: PMCID 34002087, 33633406, 38658557 and 28462073, to name a few. This literature needs to be further integrated. This is particularly relevant given the current evidence suggesting that different POMC neuronal subpopulations may not respond to peripheral metabolic and hormonal signals at all. In this context, it would also be interesting to expand the discussion on how the heterogeneity of these neurons might relate to differential control by inflammatory peripheral signals in the context of sepsis, as discussed in Section 5.

We thank the reviewer for this insightful comment. In response, we have substantially expanded the discussion of POMC neurons and incorporated this content into Section 2: “Notably, accumulating evidence indicates that POMC neurons are functionally heterogeneous, and their activation does not always oppose feeding. Under specific physiological or experimental conditions, stimulation of POMC neurons can elicit behavioural responses that resemble those induced by AgRP/NPY neurons, including the promotion of food intake (Koch *et al*, 2015). This apparent functional diversity is likely attributed to molecular and anatomical heterogeneity within the POMC neuronal population. Single-cell RNA sequencing studies have revealed distinct POMC neuron subtypes, including populations expressing low levels of POMC alongside high AgRP expression, as well as canonical POMC neurons with high POMC and minimal AgRP expression, which can be further segregated into 4 distinct clusters. Quarta *et al*. (2021) comprehensively reviewed the classification strategies used to define POMC neuron subtypes and discussed how this heterogeneity relates to appetite regulation (Quarta *et al*, 2021). In line with this heterogeneity, Biglari *et al*. (2021) demonstrated that leptin receptor expressing ($Lepr^+$) and glucagon-like peptide-1 receptor expressing ($Glp1r^+$) POMC neurons exhibit minimal anatomical overlap and exert divergent effects on feeding, with selective activation of $POMC^{Lepr^+}$ neurons producing only modest anorexia, whereas stimulation of $POMC^{Glp1r^+}$ neurons robustly promote satiety (Biglari *et al*, 2021). Furthermore, lineage tracing combined with single-cell profiling has uncovered an atypical population of POMC-lineage neurons, termed “ghost neurons”, that express negligible levels of POMC. The abundance of these cells increases in diet-induced obesity independently of neurogenesis or cell death and can be reversed by weight loss,

highlighting the dynamic nature of POMC neuron identity and function (Leon *et al*, 2024).”

Until now, no research has been published in which has been investigated what the role of these different POMC neuron subtypes is during systemic inflammation or sepsis. Therefore, we could not elaborate on this in Section 5, but have included that it would be of great interest to address how these different subpopulations of POMC neurons behave under septic conditions and whether one of these subtypes can be identified as a potential target to reverse sepsis-associated anorexia.

- The review offers a 'neurocentric view' of how inflammatory, hormonal or neuronal signals might impact AgRP or POMC neurons in the context of sepsis. However, it should be noted that sepsis and brain inflammation are also influenced by microglia, astrocytes and peripheral immune cells that can travel to the brain and perpetuate the inflammatory response. Considering the pivotal role of brain immune cells and neuroglia in POMC and AgRP neuronal biology, it is surprising that the authors did not address this issue. I believe this concept warrants integration and further discussion in the manuscript.

We fully agree with the reviewer that sepsis-associated brain inflammation is not solely mediated by peripheral inflammation, but critically involves microglia, astrocytes, and infiltrating peripheral immune cells, which together shape neuronal inflammation and influence hypothalamic neuronal function. In response to this comment, we have substantially revised the manuscript and integrated this concept in Section 4.2 and 5.1.

- It has recently been shown that hypothalamic circuits in mice can rapidly modulate gut microbiota (PMID: 40263603), which have been postulated to play a role in sepsis (see, for example, PMID: 39779878 or others). It may be interesting to expand on this potential link.

Indeed, as suggested by the reviewer, the gut microbiome also plays an important role in sepsis. Therefore, we have included a part in the main text of the review about how AgRP or POMC neurons can affect the gut microbial profile and how this might play a role in sepsis: Beyond immune and metabolic modulation, hypothalamic feeding circuits have been implicated in regulating gut microbiota composition, as activation or inhibition of AgRP or POMC neurons rapidly reshapes microbial profile (Toledo *et al*, 2025). Given the emerging role of the gut microbiome in sepsis, hypothalamic circuits might influence host responses indirectly via microbiota-dependent mechanisms. For example, Keskey *et al*. (2025) show that gut microbial metabolites, together with pathogen factors, critically affect survival in mouse models of systemic infection relevant to human sepsis (Keskey *et*

al, 2025). Whether coordinated interactions between hypothalamic feeding circuits and the gut microbiome contribute to disease progression during sepsis remains an important open question.

- It has been demonstrated that hypothalamic AgRP neurons exert top-down control over systemic TNF- α release during endotoxemia (PMID: 36182699). AgRP neurons are also both sufficient and necessary for reducing circulating Ly6H classical monocytes during fasting (PMID: 40184437). Therefore, while the authors focus on describing peripheral-to-brain axes that potentially control neural activity during sepsis, they should also consider the importance of brain-to-periphery feedback signals that regulate inflammatory responses during sepsis. Further discussion on this topic is warranted.

We thank the reviewer for highlighting the potential role of brain-to-periphery feedback in modulating immune responses during sepsis. We have now added a discussion of this concept in the revised manuscript, emphasizing that in addition to peripheral-to-brain signalling, central feeding circuits can provide top-down regulation of inflammatory responses and host defence, highlighting a bidirectional communication between the brain and the periphery during sepsis.

- Overall, the manuscript would benefit from a more in-depth analysis of how the authors can critically explain some of the contradictory findings in this field. For example, section 4.1 cites studies showing that vagotomy significantly increases mortality and LPS-induced endotoxemia, while others report the opposite, as the same procedure can apparently reduce inflammation-induced anorexia, inflammatory responses and sickness behaviour. Similarly, in Section 5, the authors report studies in which LPS prevents fasting-induced cFOS activation of AgRP neurons, alongside others showing that, in fasted mice with LPS-induced endotoxemia, AgRP neuron activity is sustained. How can these opposing lines of evidence be reconciled? Can the authors share their thoughts on these contrasting findings?

We thank the reviewer for raising this important point. In section 4.1, we have explained these contradictory results considering vagotomy and included this in the main text of the review: "These contradictory outcomes might reflect the dual and context-dependent functions of vagal nerve signalling. The vagal afferents sense peripheral inflammatory mediators and transmit these signals to the CNS, contributing to sickness behaviour and anorexia, while efferent vagal output engages the cholinergic anti-inflammatory reflex to restrain excessive cytokine release and prevents tissue injury in multiple models of systemic inflammation, shock, and sepsis (Huston, 2012). Disruption of vagal signalling

may be beneficial in attenuating maladaptive behavioural responses during mild or acute inflammatory challenges, but might be harmful in severe or systemic inflammatory states where vagal nerve-mediated anti-inflammatory control is needed. The effect of vagotomy might depend on the inflammatory context, disease severity, and the balance between afferent and efferent vagal pathways engaged. Further studies are needed to investigate how the effects of vagotomy depend on the state of inflammation and disease severity to clarify the context-dependent role of vagal signalling across septic conditions.”

For the section 5, we think these findings are not necessarily contradictory, but rather reflect differences in experimental read-outs, temporal dynamics, and functional interpretation of AgRP neuronal activity under inflammatory conditions. Therefore, we have rephrased this in the main text of the review: “Hao *et al.* (2016) have assessed AgRP neuron activation using cFos expression and have shown that LPS-induced endotoxemia blunts fasting-induced cFos activation and refeeding responses, suggesting impaired engagement of canonical hunger-related transcriptional programs (Hao *et al.*, 2016). In contrast, Su *et al.* (2017) and Boutagouga Boudjadja *et al.* (2022) directly measured neuronal activity using electrophysiological or calcium-based approaches and reported that AgRP neuron activity can be sustained in fasted mice during endotoxemia (Su *et al.*, 2017; Boutagouga Boudjadja *et al.*, 2022). These observations suggest that inflammatory states may uncouple neuronal firing from immediate early gene induction and behavioural output. Importantly, despite preserved AgRP neuronal activity, multiple studies have shown that optogenetic or chemogenetic activation of AgRP neurons fails to restore food intake following LPS administration, indicating that inflammatory signals impose circuit-level constraints downstream of AgRP neurons. Together, these findings support a model in which AgRP neuron activity is maintained or altered in inflammatory contexts but is insufficient to drive feeding behaviour, likely due to upstream inhibitory inputs and/or downstream override of feeding circuits.”

-In Section 4.2, the authors emphasise the evidence implicating tanycyte cells in the selective uptake of hormones into the hypothalamus. While the cited papers support this hypothesis, others that were not reported do not (PMCID: PMC7423758; PMID: 30941008). Therefore, this concept should be toned down.

We have taking this comment about the concept of tanycytes and hormonal transport into account and therefore we have implemented these references into the main text of the review: “On the contrary, Yoo *et al.* (2019, 2020) have demonstrated that selective

deletion of LepR in tanycytes does not affect leptin-induced pSTAT3 expression in hypothalamic neurons. Furthermore, using mice in which tanycytes in the ARC and ME are conditionally ablated, they have observed that tanycytes are not required for the active transport of leptin from the blood into the hypothalamus (Yoo *et al*, 2019, 2020).”

-section 5.2. 'It is clear that GLP-1 regulates food intake, as GLP-1 or a GLP-1RA are taken up by POMC/CART neurons by GLP-1R, which directly stimulates these neurons and indirectly inhibits AgRP/PP neurons via GABA-dependent signalling'. Please rephrase this sentence, as it is unclear in its current form.

We have rephrased this sentence: “GLP-1 is known to regulate food intake, at least in part through hypothalamic circuits. Evidence suggests that GLP-1 or a GLP-1RA can directly stimulate POMC/CART neurons by binding to the GLP-1R, while indirectly suppressing AgRP/NPY neuron activity *via* GABA-dependent signalling.”

REVIEWER 2

This is an interesting and relatively comprehensive review of potential mediators of sepsis-associated anorexia, focusing on the hypothalamus. It is generally well-written and contains several figures, most of which are informative.

We thank the reviewer for these kind words.

Overall, while it is understood that this review focuses on potential hypothalamic mechanisms, there has been a great deal of recent progress in the understanding of roles of the area postrema and nucleus of the solitary tract in this process. It would seem useful to mention these results, if only to place them in context of the hypothalamic circuits.

We thank the reviewer for this thoughtful suggestion. We agree that recent advances regarding the roles of the AP and NTS in the regulation of feeding and sickness behaviours are highly interesting and relevant. However, due to space limitations and the primary focus of this review on sepsis-related hypothalamic mechanisms, we have limited the expansion of this section. Instead, we have incorporated key findings related to infection and sepsis where appropriate and referred readers to the comprehensive review by Johansen *et al*. (2025) for detailed background information on NTS and AP (Johansen *et al*, 2025). We believe this approach allows us to place brainstem mechanisms in proper context while maintaining the scope and focus of the present review. This aspect was

included in section 4.1 of the review.

Some of the subsections contain a great deal of information, but little synthesis. It could be useful to include concluding sentences at the end of each section. While this is true of many subsections, sections 4.1 and 4.2 represent examples of this.

We agree that addressing this point strengthens the logical flow of the manuscript. In response, we have added brief summaries and conclusions to each subsection, which are highlighted in yellow in the revised manuscript.

Figure 3 is very busy and confusing. One potential way to simplify it could be to note that the Bnc2, Trh, DMH Glp1r, and Lepr/Glp1r neurons all represent the same population of neurons that mediate the inhibition of Agrp neurons. It could also be useful to note the putative functions of the second order neurons.

We have revised Figure 3 to improve clarity and readability. Specifically, we simplified the schematic by removing redundant arrows and visual elements. We have annotated the putative functions of the second-order neurons directly in the figure. We believe that these changes substantially reduce visual complexity and make the signalling relationships easier to interpret.

Minor remarks:

Page 5- THR, CRH, and OXT neurons of the PVH have been shown to play little role in the suppression of food intake.

We agree with this. Thank you for pointing it out. They are considered as the candidate to promote negative energy balance. Thus, we will not include this into the text.

Last sentence of the top paragraph on page 7- "completely reversed" might make more sense than "completely restored"

We adjust "completely restored" into "completely reversed".

Page 8- the discussion of GRB10 seems out of place in a paragraph that otherwise focuses on AGF/SPX. Perhaps it would make more sense to discuss GRB10 in the context of SHP2 (to which GRB10 binds) on page 7.

We thank the reviewer for the suggestion regarding the placement of the GRB10 discussion. After thorough literature review, we did not find direct experimental evidence demonstrating a physical interaction between GRB10 and SHP2. Furthermore, the purpose

of this section is to discuss how these factors influence POMC neuron activity via the leptin signalling pathway. Therefore, we believe that keeping the GRB10 discussion in this context is appropriate.

REVIEWER 3

The review is timely and is potentially of broad interest. It does a nice job of summarizing many of the key anorexigenic pathways. However, in its current form it is lacking in several areas. Several suggestions are below.

We thank the reviewer for these kind words.

The entire crux of the article seems to be that better understanding of anorexia during sepsis may lead to potential therapies to treat sepsis-associated anorexia. The entire review seems to be focused on the hypothesis that sepsis-induced anorexia is detrimental and should be reversed. However, they do not provide any evidence to support this hypothesis. They mention in the introduction that critically ill patients do not benefit from early full feeding but then never expand on this.

We thank the reviewer for raising this important point. We agree that in the original version of the manuscript, the clinical evidence related to nutritional strategies in sepsis, particularly the timing and extent of feeding, was not sufficiently discussed. To address this concern, we have now added a dedicated section summarizing recent clinical studies and meta-analyses examining enteral nutrition, permissive underfeeding, and phase-specific energy provision in patients with sepsis.

Moreover, the authors could make a better case for why anorexia during sepsis should even warrant treatment. Perhaps an entire section discussing the pros and cons of sepsis-associated anorexia. They could then describe complications such as muscle catabolism, immune suppression, etc. to support their potential treatment strategies.

We agree that it is important to clarify why sepsis-associated anorexia may warrant therapeutic consideration. In the revised manuscript, we have strengthened this aspect by incorporating evidence from clinical observations indicating that they could benefit from nutritional treatment. At the same time, we acknowledge that sepsis-associated anorexia may also exert context-dependent protective effects, particularly during certain infectious conditions. Given space limitations and the scope of the current review, we did not include

a comprehensive discussion of the pros and cons of sepsis-associated anorexia. Notably, the potential beneficial and detrimental roles of sickness-induced anorexia have already been thoroughly discussed in several recent, high-quality reviews, which we now explicitly cite. Our review aims to complement these works by focusing on mechanisms that may underlie maladaptive or prolonged anorexia in sepsis and by highlighting potential targets for therapeutic intervention. We hope that this revised framing clarifies our intent and better justifies the relevance of targeting anorexia in sepsis within a mechanistic and translational context.

The article does not explicitly discuss the anti-inflammatory actions of alpha-MSH. It may be worth adding in discussion of how an anti-inflammatory peptide is linked to anorexia. They discuss alpha-MSH only as a "satiety" peptide. Similarly, a discussion on the potential anti-inflammatory effects of ghrelin seems missing.

We agree with the reviewer that discussing the anti-inflammatory potential of α -MSH, ghrelin, and GLP-1 together with the fact that leptin acts more as a pro-inflammatory mediator might be interesting. However, due to space limitations and the focus of our review, we were not able to include this into the manuscript.

The authors could include a discussion of central GLP1 signalling inhibits TLR4 signalling. Given the broad use of GLP-1 drugs, this seems relevant.

Also here, this is a valid point from the reviewer. Similar to the previous comment, we believe this would go too far beyond the focus of our manuscript and therefore we did not include this into the revised version.

The authors should also recently work by the Dulac group regarding a preoptic neuronal population controlling fever and appetite during sickness.

We thank the reviewer for mentioning this interesting study. As we focus on the AgRP/POMC circuit in feeding behaviour, we have now incorporated a discussion of the relevant findings from the Dulac group. Specifically, we discuss evidence that LPS activates neurons in the ventromedial preoptic area (VMPO) of the hypothalamus, and that these LPS-responsive VMPO neurons induce anorexia by directly inhibiting projections to AgRP neurons. In addition, these neurons may indirectly activate POMC and CART neurons in the ARC, thereby further contributing to sickness induced anorexia. This discussion has been added to page 17.

Although this review is focused on hypothalamus, the Friedman lab identified a key set of

brainstem neurons control multiple aspects of sickness behaviour.

We thank the reviewer for highlighting this important study. We agree that the identification of a set of neurons in brainstem especially the ADCYAP1 neurons by the Friedman lab provides critical insights into the neural control of sickness behaviour. We have now added this work to the revised manuscript and integrated it into the broader framework, emphasizing its potential interaction with hypothalamic circuits in regulating sickness associated behaviour. We have included this in section 4.1, page 15.

Page 3, the sentence "Furthermore, long-term anorexia during sepsis will lead to insufficient energy intake, which is correlated with poor patient outcomes (Kitayama et al 2023). This sentence does not properly reflect this study, which examined the presence of poor appetite 12 months after discharge from ICU (only 10% of admissions were for sepsis), and presence of malignancy at time of ICU admission seems to be driving the vast majority of persistent poor appetite 12 months later.

Thanks for pointing this out. We have reorganized this part of the manuscript and combined it with the suggestion of why to induce feeding during sepsis.

The authors comment on studies of sickness-associated anorexia as an important characteristic of both disease resistance and tolerance during sepsis but don't discuss the work from the Mediative group.

We have included this reference to the general introduction of the revised manuscript.

References

- Biglari N, Gaziano I, Schumacher J, Radermacher J, Paeger L, Klemm P, Chen W, Corneliussen S, Wunderlich CM, Sue M, *et al* (2021) Functionally distinct POMC-expressing neuron subpopulations in hypothalamus revealed by intersectional targeting. *Nat Neurosci* 24: 913–929
- Boutagouga Boudjadja M, Culotta I, De Paula GC, Harno E, Hunter J, Cavalcanti-de-Albuquerque JP, Luckman SM, Hepworth M, White A, Aviello G, *et al* (2022) Hypothalamic AgRP neurons exert top-down control on systemic TNF- α release during endotoxemia. *Curr Biol* 32: 4699-4706.e4
- Hao L, Sheng Z, Potian J, Deak A, Rohowsky-Kochan C & Routh VH (2016) Lipopolysaccharide (LPS) and tumor necrosis factor alpha (TNF α) blunt the response of Neuropeptide Y/Agouti-related peptide (NPY/AgRP) glucose inhibited (GI) neurons to decreased glucose. *Brain Res* 1648: 181–192
- Huston JM (2012) The vagus nerve and the inflammatory reflex: wandering on a new treatment paradigm for systemic inflammation and sepsis. *Surg Infect (Larchmt)* 13: 187–93
- Johansen VBI, Petersen J, Lund J, Mathiesen CV, Fenselau H & Clemmensen C (2025) Brain control of energy homeostasis: Implications for anti-obesity pharmacotherapy. *Cell* 188: 4178–4212
- Keskey RC, Xiao J, Hyoju S, Lam A, Kim D, Sidebottom AM, Zaborin A, Dijkstra A, Meltzer R, Thakur A, *et al* (2025) Enterobactin inhibits microbiota-dependent activation of AhR to promote bacterial sepsis in mice. *Nat Microbiol* 10: 388–404
- Koch M, Varela L, Kim JG, Kim JD, Hernández-Nuño F, Simonds SE, Castorena CM, Vianna CR, Elmquist JK, Morozov YM, *et al* (2015) Hypothalamic POMC neurons promote cannabinoid-induced feeding. *Nature* 519: 45–50
- Leon S, Simon V, Lee TH, Steuernagel L, Clark S, Biglari N, Lesté-Lasserre T, Dupuy N, Cannich A, Bellocchio L, *et al* (2024) Single cell tracing of Pomc neurons reveals recruitment of ‘Ghost’ subtypes with atypical identity in a mouse model of obesity. *Nat Commun* 15: 3443
- Quarta C, Claret M, Zeltser LM, Williams KW, Yeo GSH, Tschöp MH, Diano S, Brüning JC & Cota D (2021) POMC neuronal heterogeneity in energy balance and beyond: an

integrated view. *Nat Metab* 3: 299–308

Su Z, Alhadeff AL & Betley JN (2017) Nutritive, Post-ingestive Signals Are the Primary Regulators of AgRP Neuron Activity. *Cell Rep* 21: 2724–2736

Toledo M, Martínez-Martínez S, Van Hul M, Laudo B, Eyre E, Pelicaen R, Puel A, Altirriba J, Gómez-Valadés AG, Inderhees J, *et al* (2025) Rapid modulation of gut microbiota composition by hypothalamic circuits in mice. *Nat Metab* 7: 1123–1135

Yoo S, Cha D, Kim DW, Hoang T V & Blackshaw S (2019) Tanycyte-Independent Control of Hypothalamic Leptin Signaling. *Front Neurosci* 13: 240

Yoo S, Cha D, Kim S, Jiang L, Cooke P, Adebessin M, Wolfe A, Riddle R, Aja S & Blackshaw S (2020) Tanycyte ablation in the arcuate nucleus and median eminence increases obesity susceptibility by increasing body fat content in male mice. *Glia* 68: 1987–2000

20th Feb 2026

Dear Dr. Vanderhaeghen,

Thank you for the submission of your revised manuscript to EMBO Molecular Medicine, and please accept my apologies for the delay in getting back to you as one referee needed more time to complete their review. We have now received the feedback from the referees, and as you will see below, they are satisfied with the revisions and support publication.

I am thus pleased to inform you that your manuscript is accepted for publication at EMM. It will be sent to our publisher once the figures will have been redrawn by our scientific illustrator and approved by you.

Your manuscript will be processed for publication by EMBO Press. It will be copy edited and you will receive page proofs prior to publication. Please note that you will be contacted by Springer Nature Author Services to complete licensing information. This Review article is free of charge.

If you have any questions, please do not hesitate to contact the Editorial Office. Thank you for your contribution to EMBO Molecular Medicine!

With kind regards,

Lise Roth

Referee #1 (Remarks for Author):

The authors have made significant improvements in the revised version of the review. It is enriched with new information and provides a more comprehensive and clearer overview of the topic. I have no further comments.

Referee #2 (Remarks for Author):

This revised review manuscript remains thorough, interesting, and timely. It has been improved by the revisions made by the authors.

Referee #3 (Remarks for Author):

The authors have done a nice job in responding to the previous critiques.